DOI: 10.1038/s41467-018-06811-z　　**OPEN**

# Heterozygous deletion of chromosome 17p renders prostate cancer vulnerable to inhibition of RNA polymerase II

Yujing Li[1,2,3], Yunhua Liu [2,3,4], Hanchen Xu[1,2,3], Guanglong Jiang[3], Kevin Van der Jeught [2,3], Yuanzhang Fang[2,3], Zhuolong Zhou[3], Lu Zhang[2,3], Michael Frieden[3], Lifei Wang[3], Zhenhua Luo[5], Milan Radovich[4,6], Bryan P. Schneider[7], Yibin Deng[8], Yunlong Liu[3], Kun Huang[7], Bin He[9], Jin Wang [10], Xiaoming He [11,12], Xinna Zhang[3,4], Guang Ji [1] & Xiongbin Lu[2,3,4]

Heterozygous deletion of chromosome 17p (17p) is one of the most frequent genomic events in human cancers. Beyond the tumor suppressor *TP53*, the *POLR2A* gene encoding the catalytic subunit of RNA polymerase II (RNAP2) is also included in a ~20-megabase deletion region of 17p in 63% of metastatic castration-resistant prostate cancer (CRPC). Using a focused CRISPR-Cas9 screen, we discovered that heterozygous loss of 17p confers a selective dependence of CRPC cells on the ubiquitin E3 ligase Ring-Box 1 (RBX1). RBX1 activates POLR2A by the K63-linked ubiquitination and thus elevates the RNAP2-mediated mRNA synthesis. Combined inhibition of RNAP2 and RBX1 profoundly suppress the growth of CRPC in a synergistic manner, which potentiates the therapeutic effectivity of the RNAP2 inhibitor, α-amanitin-based antibody drug conjugate (ADC). Given the limited therapeutic options for CRPC, our findings identify RBX1 as a potentially therapeutic target for treating human CRPC harboring heterozygous deletion of 17p.

[1] Institute of Digestive Diseases, Longhua Hospital, Shanghai University of Traditional Chinese Medicine, 200032 Shanghai, China. [2] Department of Cancer Biology, The University of Texas MD Anderson Cancer Center, Houston, TX 77030, USA. [3] Department of Medical and Molecular Genetics, Indiana University School of Medicine, Indianapolis, IN 46202, USA. [4] Indiana University Melvin and Bren Simon Cancer Center, Indiana University School of Medicine, Indianapolis, IN 46202, USA. [5] The Liver Care Center and Division of Gastroenterology, Hepatology and Nutrition, Cincinnati Children's Hospital Medical Center, Cincinnati, OH 45229, USA. [6] Department of Surgery, Indiana University School of Medicine, Indianapolis, IN 46202, USA. [7] Department of Medicine, Indiana University School of Medicine, Indianapolis, IN 46202, USA. [8] Laboratory of Cancer Genetics, The University of Minnesota Hormel Institute, Austin, MN 55912, USA. [9] Biomarker Research Program Center, Houston Methodist Research Institute, Houston, TX 77030, USA. [10] Department of Pharmacology and Chemical Biology, Baylor College of Medicine, Houston, TX 77030, USA. [11] Fischell Department of Bioengineering, University of Maryland, College Park, MD 20742, USA. [12] Marlene and Stewart Greenebaum Comprehensive Cancer Center, University of Maryland, Baltimore, MD 21201, USA. These authors contributed equally: Yujing Li, Yunhua Liu, Hanchen Xu. Correspondence and requests for materials should be addressed to X.Z. (email: xz48@iu.edu) or to G.J. (email: jiliver@vip.sina.com) or to X.L. (email: xiolu@iu.edu)

Prostate cancer is among the most common male malignancies and one major leading cause of cancer mortality in men[1,2]. Since the discovery of androgen dependence in prostate cancer, the backbone therapy for prostate cancer has been to lower androgen levels by surgical castration or androgen-deprivation therapy[3]. However, although many patients initially respond to androgen deprivation therapy, nearly all the patients relapse and eventually develop castration-resistant prostate cancer (CRPC)[4]. Over the past decade, it has become clear that the androgen receptor (AR) plays a pivotal role in the development of resistance to hormone therapies in both primary and recurrent prostate cancer. New therapeutic approaches in advanced prostate cancer have focused on the AR protein, which led to the development of AR-targeting agents, abiraterone acetate and enzalutamide[5].

Despite the success of androgen deprivation and AR-blocking therapies, the newly developed therapies also suffer a short therapeutic durability due to acquired resistance[4,6]. Thus, researchers are now searching for more therapeutic targets, one of which is prostate-specific membrane antigen (PSMA). PSMA is highly expressed on the surface of nearly all prostate cancer cells but is present on only a few normal tissues, making it an excellent target for drugs that selectively attack tumors[7]. Radioactive element lutetium-177-labeled PSMA antibody has shown promise in treating patients who are resistant to other drug therapies[8–10]. While it is hard to directly target variant forms of AR or alterations in the AR gene that promote castration resistance, a small molecular inhibitor against ROR-γ, an upstream regulator of AR, was proven to effectively shut down AR signaling[11]. In mouse CRPC models, treatment with ROR-γ inhibitors led to substantial and prolonged regression of tumors, and restored their sensitivity to the treatment of enzalutamide. In-depth understanding of prostate cancer invasion, metastasis and drug resistance will help identify more therapeutic targets and lead to new treatment options.

The whole-genome sequencing of cancer genomes and other associated omics efforts have empowered our knowledge of human cancer biology and pathogenesis. These efforts have facilitated personalized medicine to find new genetic alterations in the context of a specific cancer. Comprehensive analyses of key genomic changes in prostate cancer will accelerate our progress in developing more effective ways to diagnose, treat and prevent this disease. Recent studies have identified recurrent somatic mutations, copy number variations, and chromosomal rearrangements in prostate cancer[12,13]. A number of frequent genomic changes are shared by primary and metastatic prostate cancer, including E26 transformation-specific (ETS) fusions, point mutations in SPOP, FOXA1, and TP53, and copy number alterations involving MYC, RB1, and PTEN, although these alterations are differentially enriched at different stages of prostate cancer[12,13]. Interestingly, aberrations of BRCA2, BRCA1, and ATM were observed at a much higher frequency in CRPC, indicating the potential application of PARP inhibitors in treating this subset of cancers. In a phase II study of olaparib[14], a PARP inhibitor, it was proven to have high response rates in patients with metastatic CRPC carrying DNA repair defects. While certain prostate cancer alterations or signatures have prognostic clinical significance, the therapeutic approach targeting those genomic events has not yet been developed.

It has been long known from cytogenetic and loss of heterozygosity (LOH) studies that deletions on the 17p frequently occur in many types of human cancer[15–17]. While TP53 loss may play a dominant role in the tumor initiation or progression, it remains unclear whether many genes in the deletion region impact tumorigenesis beyond TP53 loss alone. A recent study showed that loss of Eif5a and Alox15b in the mouse 11B3 (syntenic to

human 17p13.1) cooperates with Trp53 (mouse orthologue of TP53) deletion to produce more aggressive disease in lymphoma and leukemia[18]. In this study, we find that 63% of metastatic prostate cancer harbors heterozygous deletion of a region that spans up to 20 megabases of DNA at 17p. We demonstrate that POLR2A is included in the 17p deletion region along with TP53 in a majority of prostate cancers. POLR2A is the catalytic subunit of the RNAP2 complex that is solely in charge of mRNA synthesis in cells. POLR2A and RNAP2 activity is specifically inhibited by α-amanitin, a small molecule peptide produced by the death cap mushroom (Amanita phalloides)[19,20]. Inhibition of POLR2A with α-amanitin-based ADC selectively suppresses the proliferation, survival and tumor growth of CRPC cells harboring this genomic event. Using a CRISPR-Cas9 screen, we uncover that heterozygous deletion of 17p confers a selective dependence on RBX1, inhibition of which had a synergistic and robust suppression in the growth of CRPC along with the treatment of α-amanitin-conjugated anti-EpCAM antibodies.

## Results

**Heterozygous deletion of 17p is frequent in prostate cancer**. TP53, a well-documented tumor suppressor gene, is frequently inactivated by mutation or deletion in a majority of human tumors[21]. However, the mutation and homozygous deletion of TP53 are relatively infrequent events in prostate cancer, accounting for only 12% (59 out of 492) and 8% (37 out of 492) of total cases, respectively. In contrast, approximately 26% (126 out of 492) of all prostate tumor samples harbor heterozygous loss of TP53 (Fig. 1a). Our recent studies have shown that heterozygous deletion of TP53 in human colorectal cancer frequently encompasses a neighboring essential gene POLR2A, rendering cancer cells with heterozygous loss of TP53 susceptible to further inhibition of POLR2A[22,23]. The comprehensive analysis of prostate cancer genomes revealed that the TP53 deletion is often included in a large fragment deletion that spans over almost the whole short arm of chromosome 17 (17p) (Fig. 1b). The loss of 17p is frequently detected at relatively late stages of colorectal cancer development[24,25], which may contribute to or even determine the transition from the early-stage to advanced-stage cancer. Consistent with the findings in colorectal cancer, analysis of The Cancer Genome Atlas (TCGA) prostate dataset revealed that 17p loss is readily detectable in low-grade T1 tumors, but was significantly increased in the advanced T2-T4 tumors (Fig. 1c, d). Moreover, the loss of 17p rises up to 63% (95 out of 150) in metastatic prostate cancer, whereas TP53 mutation and homozygous deletion account for only 39 and 6% respectively (Fig. 1e, f). Collectively, these results indicate the 17p loss may play an important role in prostate cancer progression and metastasis.

**RBX1 is an essential gene for 17p^loss CRPC cells**. In the 17p deletion region, there are as many as 200 protein-coding genes and noncoding RNA genes, as well as even more regulatory elements for transcriptional and epigenetic activities. To identify genes for which loss of function would lead to selective killing of 17p-deficient cells, we performed a CRISPR-based genetic screen in DU145 (a CRPC cell line) and its isogenic line that had been genetically modified to delete one copy of 17p from WDR81 to MAP2K3 with the size of ~19.6 megabases (Supplementary Fig. 1a–d). The isogenic pair of 17p^loss and 17p^neutral DU145 cells displayed similar proliferation rate, suggesting that single 17p is sufficient to maintain cell viability and proliferation (Supplementary Fig. 1e). A pool of lentiviral CRISPR single guide RNA (sgRNA) library was engineered to express sgRNAs targeting 3733 genes whose protein products localize mainly in the nucleus

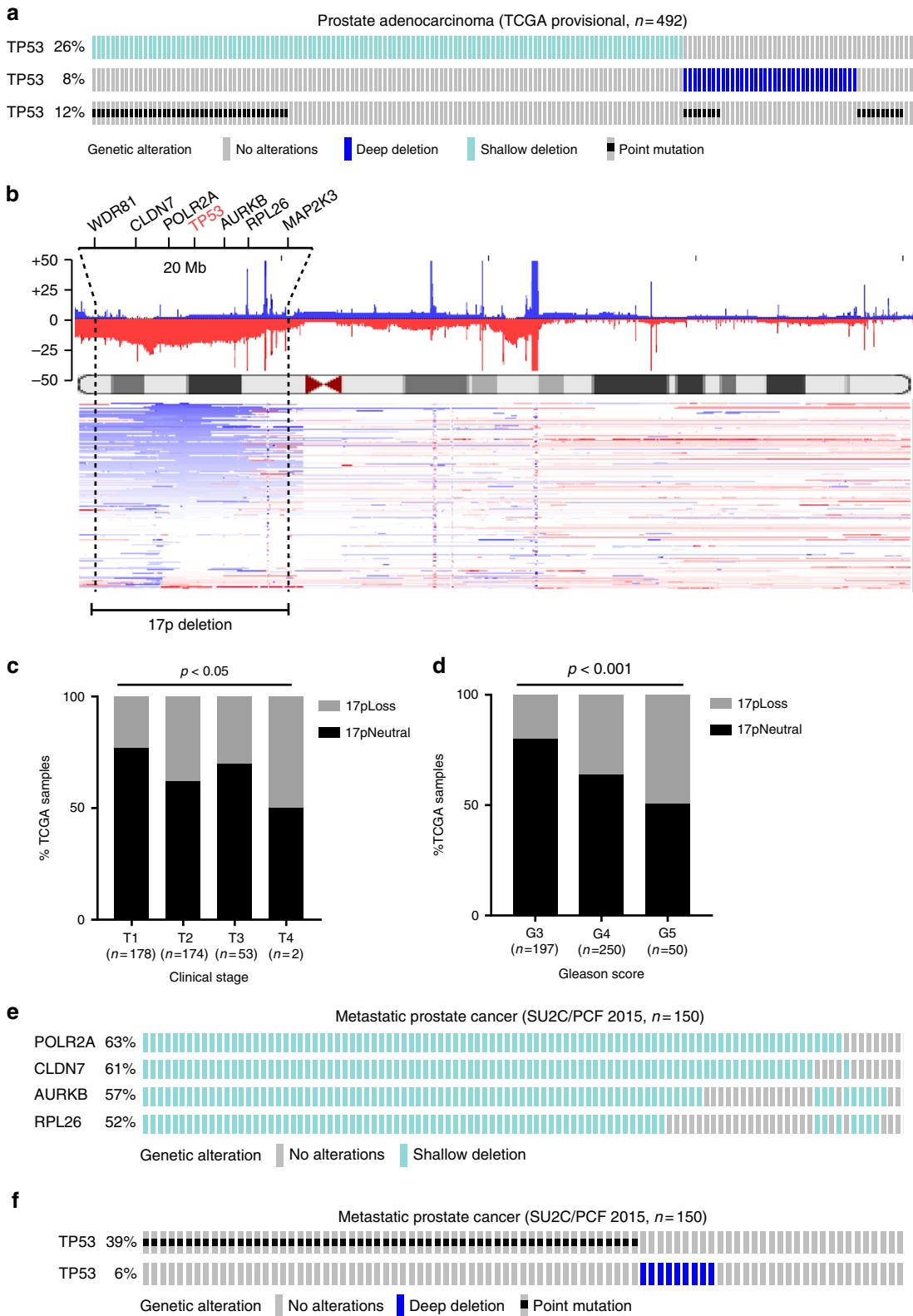

**Fig. 1** Chromosome 17p loss is a frequent genomic event in prostate cancer. **a** Genomic alterations of *TP53* (point mutation, shallow deletion and deep deletion) in a TCGA prostate cancer dataset (TCGA provisional, *n* = 492) determined by cBioportal. Due to the intra-tumor heterogeneity, one tumor tissue may include tumor cells with homozygous deletion of TP53 or with mutant TP53, as shown in a few cases. **b** Integrated analysis of 17p deletion in 155 prostate cancer patient samples. Frequency plots of the copy number abnormalities indicate degree of copy number loss (red) or gain (blue). Representative genes in 17p deletion region are shown. **c**, **d** Distribution of heterozygous deletion of 17p among tumor clinical (**c**) and pathological (**d**) stages in the TCGA prostate cancer dataset (Fisher's exact test). **e**, **f** Genomic alterations of 17p heterozygous deletion (**e**) and *TP53* mutations (**f**) in the TCGA metastatic prostate cancer dataset (SU2C/PCF 2015, *n* = 150) determined by cBioportal

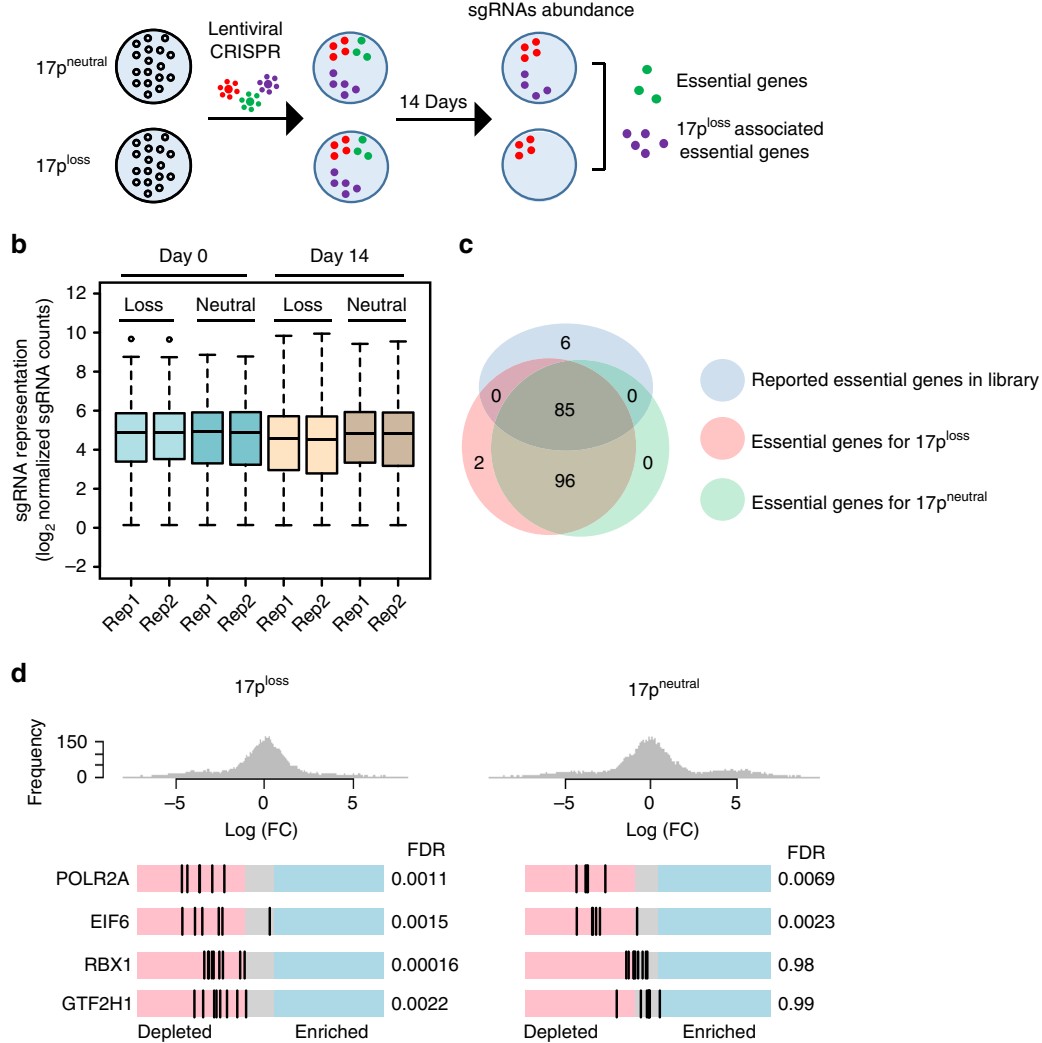

**Fig. 2** CRISPR-Cas9 screen identifies RBX1 as an essential gene for 17p$^{loss}$ prostate cancer cells. **a** Schematic illustration of CRISPR screening procedure in the isogenic pair of DU145 cells. **b** Box plots showing the distribution of sgRNA frequencies at different time points. **c** Overlapping of essential genes from this screen and from the previous reports. **d** Frequency histograms of enriched or depleted sgRNAs. *POLR2A* and *EIF6* are two representatives of common essential genes. RBX1 and GTF2H1 are representatives of selective essential genes in the context of 17p loss

(10 sgRNAs per gene, 37,330 sgRNAs in total). We infected the isogenic pair of DU145 cell lines with the pooled lentiviruses, and then assessed the relative depletion of sgRNAs after 14 population doublings in order to identify genes whose depletion significantly affected the proliferation of the 17p$^{loss}$ cells versus the 17p$^{neutral}$ cells in vitro (Fig. 2a). High correlations were achieved among replicates (Fig. 2b). Based on differential sgRNA representation with empirical analysis of digital gene expression data in R (edgeR)[26], we uncovered a total of 181 essential genes (false-discovery rate (FDR) < 0.05) that are independent of 17p loss in DU145 cells. In comparison with available databases[27,28], our screen recovered 93.4% of the known core essential genes (85 of 91) included in the library and discovered 96 essential genes specific to DU145 cells, suggesting the reliability of our screen (Fig. 2c and Supplementary Data 1). We next sought to identify essential genes only in the context of 17p$^{loss}$. To this end, the criterion is that the FDR of the gene in the 17p$^{loss}$ cells should be less than 0.05, but be more than 0.9 in the 17p$^{neutral}$ cells. Two genes (*RBX1* and *GTF2H1*) were identified to meet this criterion (Fig. 2d).

**17p$^{loss}$ CRPC cells are sensitive to RBX1 depletion**. To validate the context-dependent essentiality of RBX1 and GTF2H1, parental and the 17p$^{loss}$ DU145 cell lines were transduced with lentivirus co-expressing red fluorescence protein (RFP) and small hairpin RNA (shRNA) against RBX1 or GTF2H1, and cell viability was monitored by the RFP signal. When RBX1 was knocked down, the RFP-positive cells were dramatically reduced in the 17p$^{loss}$ cells 14 days post lentiviral transduction, but not in the parental 17p$^{neutral}$ cells (Fig. 3a, b and Supplementary Fig. 2a). Depletion of GTF2H1 also showed preferable cell killing effect in the 17p$^{loss}$ DU145 cells (Supplementary Fig. 2b-d). Recent studies identified RBX1 as a potentially druggable target for cancer therapeutics[29,30]. Therefore, we focused on RBX1 for further investigation. To confirm the context-dependent essentiality of RBX1, we generated stable cell lines expressing doxycycline (Dox)-inducible RBX1 shRNA in a panel of 17p$^{neutral}$ (22Rv1 and DU145) and 17p$^{loss}$ (PC3 and VCaP) cells. 22Rv1, DU145, and PC3 are CRPC cell lines, while VCaP is not a CRPC line but harbors 17p loss. Specifically, the expression of Dox-induced RBX1 shRNA led to markedly reduced proliferation in 17p$^{loss}$

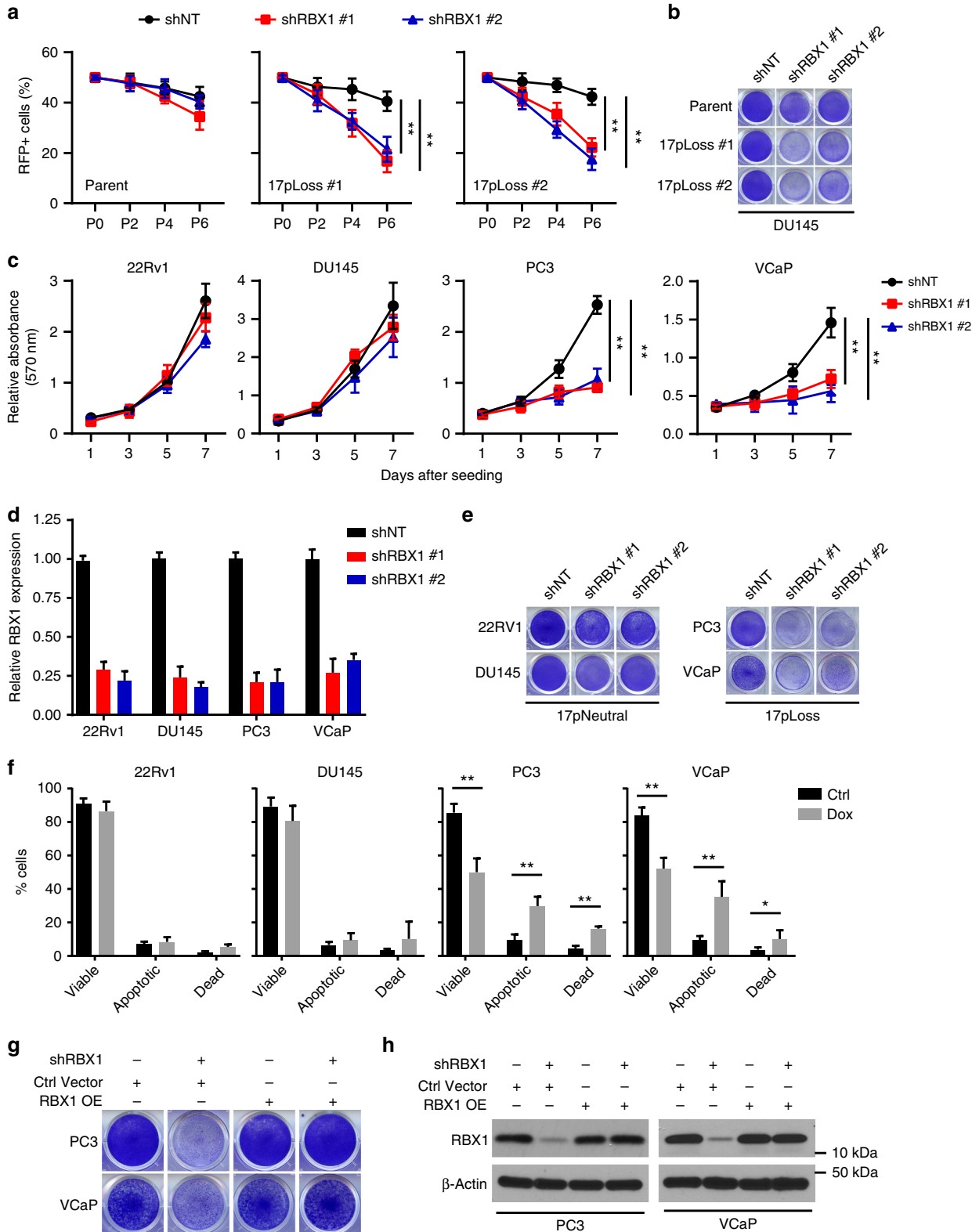

cells, in comparison with that of the corresponding cells expressing control shRNA (Fig. 3c–e). Despite significant knockdown of RBX1, the 17p$^{neutral}$ cells (22Rv1 and DU145) continued to proliferate, whereas the 17p$^{loss}$ cells (PC3 and VCaP) exhibited severe apoptosis (Fig. 3f). These results were in agreement with our genomic screen. This growth defect was rescued by introduction of a shRNA-resistant RBX1 cDNA construct in the 17p$^{loss}$ PC3 and VCaP cells, indicating the on-target effect of the

shRNA (Fig. 3g, h). The results suggest that RBX1 is a selectively essential factor in prostate cancer cells harboring 17p loss.

**Depletion of RBX1 inhibits 17p$^{loss}$ CRPC tumor growth in vivo.** To investigate the effect of RBX1 inhibition in CRPC tumors, we conducted xenograft tumor studies using parental or isogenic 17p$^{loss}$ DU145 cells expressing Dox-inducible

**Fig. 3** Prostate cancer cells with 17p deletion are highly sensitive to RBX1 depletion. **a, b** Effect of RBX1 knockdown on the proliferation of the parental and isogenic 17p$^{loss}$ DU145 cells, determined by direct competition assay. Cells expressing RFP and control nonspecific shRNA (shNT) or shRBX1 were sorted and mixed with control RFP-negative cells (1:1) and the RFP-positive cells were quantified at passages 2, 4, and 6 (**a**). Representative cell survival measured by staining with crystal violet was shown in **b**. **c–e** Cell growth curves, based on crystal violet staining, of human prostate cancer cell lines expressing Dox-inducible shNT or RBX1-specific shRNA (shRBX1 #1, shRBX1 #2) (**c**). RBX1 knockdown efficiency and representative image were shown in **d** and **e**, respectively. **f** Fraction of apoptotic cells in the 17p$^{neutral}$ (22Rv1 and DU145) and 17p$^{loss}$ (PC3 and VCaP) cells expressing Dox-induced RBX1 shRNA at 4 days post Dox treatment. **g, h** Cell survival measured by crystal violet staining (**g**) and protein expression levels (**h**) of RBX1 in PC3 and VCaP cells expressing shRBX1, control or ectopic RBX1. Data are representative of three independent experiments and analyze by unpaired two-tailed $t$-test. Error bars denote SD. **, $p < 0.01$; ***, $p < 0.001$

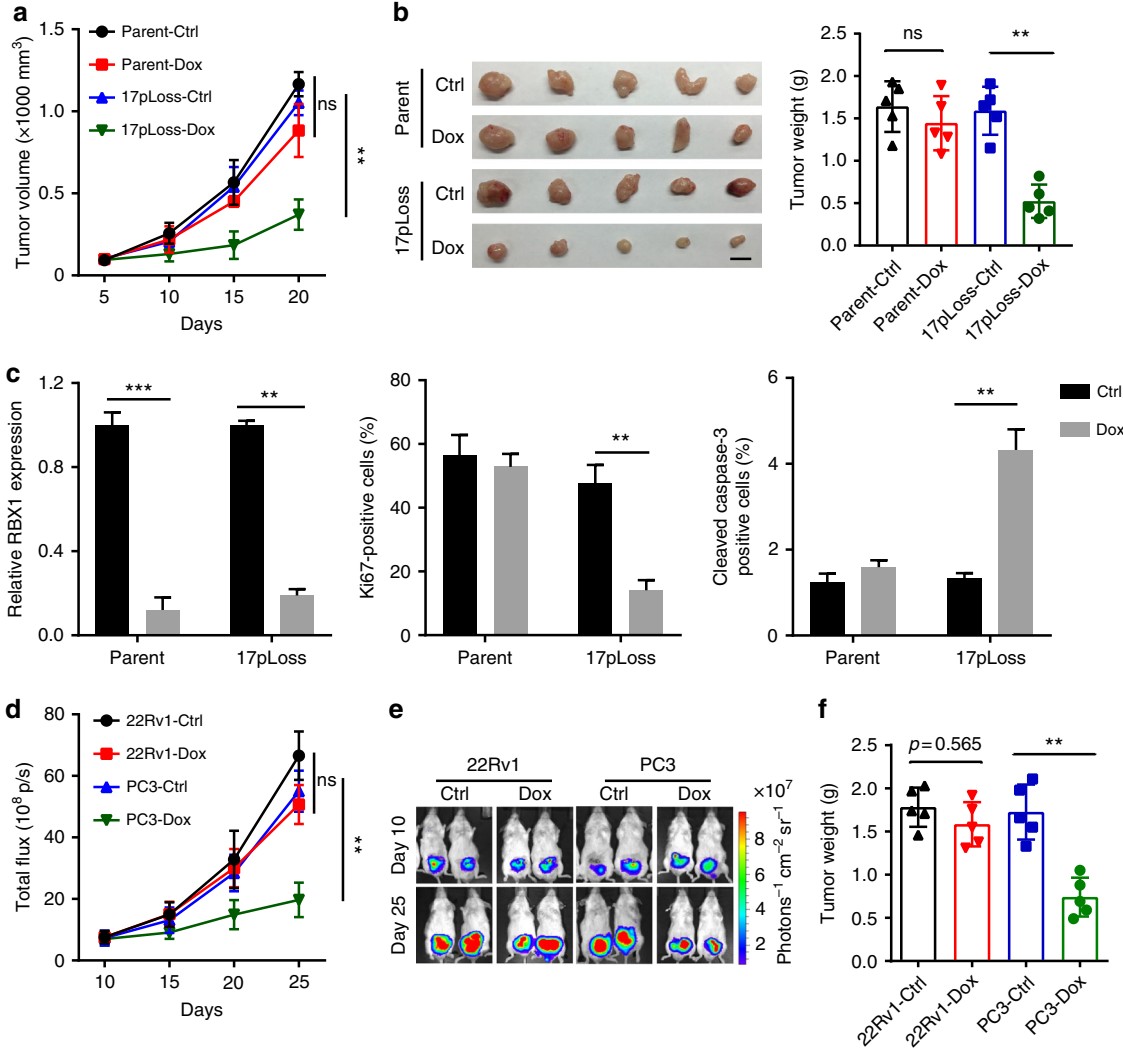

**Fig. 4** Depletion of RBX1 inhibits the growth of 17p$^{loss}$ CRPC tumors in vivo. **a–c** Tumor growth curves (**a**), gross tumor images and weights (**b**) of xenograft tumors derived from subcutaneously implanted parental and isogenic 17p$^{loss}$ DU145 cells expressing Dox-inducible RBX1 shRNA ($n = 3$). **c** RBX1 expression, cell proliferation and apoptosis in the above xenograft tumors were quantified. Scale bar, 10 mm. **d–f** Tumor growth curves (**d**), representative bioluminescent images (**e**), and gross tumor weights (**f**) of xenograft tumors derived from orthotopically implanted 22Rv1 and PC3 cells expressing Dox-inducible RBX1 shRNA ($n = 5$). Data are analyze by unpaired two-tailed $t$-test and are presented as the mean ± SD. ns not significant; **, $p < 0.01$

RBX1 shRNA. In these studies, subcutaneously implanted tumors were allowed to grow before Dox treatment for ~7 days. In accordance with our findings in vitro, administration of mice with doxycycline food markedly decreased the growth of xenograft tumors derived from the 17p$^{loss}$ DU145 cells (Fig. 4a, b). Efficiency of RBX1 knockdown in vivo was confirmed by immunohistochemistry (IHC) staining (Fig. 4c and Supplementary Fig. 3a). However, there was no substantial difference

between Dox-treated and -untreated tumors derived from the parental DU145 cells (Fig. 4a, b). Examination of end-point tumor burden in Dox-treated or -untreated groups ($n = 5$ per group) demonstrated that depletion of RBX1 led to profound decreases in tumor weight (68.5% reduction, $p < 0.001$, Fig. 4b) of the 17p$^{loss}$ tumors, and correspondingly, these tumors had a marked reduction in cell proliferation (as measured by Ki-67 levels) and a significant increase in cell apoptosis (as measured by

cleaved caspase-3 levels) (Fig. 4c and Supplementary Fig. 3a). We next sought to assess sensitivity to RBX1 ablation in an orthotopic tumor model using 22Rv1 cells (17p$^{neutral}$) and PC3 cells (17p$^{loss}$). NOD/SCID mice were injected orthotopically with 22Rv1 or PC3 cells into the dorsal lobe of prostate in order to establish prostate tumor models. In line with observations made in the DU145 xenograft mouse model, Dox-induced knockdown of RBX1 led to a robust reduction in the growth of the 17p$^{loss}$ PC3 tumors (Fig. 4d–f and Supplementary Fig. 3b). However, RBX1 depletion only had a modest effect on the growth of the 17p$^{neutral}$ 22Rv1 tumors, suggesting that targeting RBX1 confers therapeutic selectivity on the 17p$^{loss}$ CRPC tumors.

**RBX1 modifies POLR2A by the K63-linked ubiquitination.** RBX1 is an E3 ubiquitin ligase and a functional component of multiple cullin-RING-based E3 ubiquitin ligase (CRL) complexes that mediate the ubiquitination and subsequent proteasomal degradation of target proteins[31–33]. To search for the direct functional association of RBX1 with the genes located in the 17p deletion region, we first performed mass spectrometry to analyze the RBX1-containing protein complex. As expected, regular components of CRLs were identified, including cullins (CUL1, CUL2, and CUL3) and those proteins for substrate recognition (SKP2, VHL). Interestingly, p53 and POLR2A were both identified as putative RBX1-interacting proteins (Supplementary Fig. 4a and Supplementary Data 2). It was noted that the 17p$^{loss}$ DU145 cells contain an inactivating mutation of the *TP53* gene on the remaining allele[34]. Thus, it is unlikely that inhibition of RBX1 kills cells by modulating p53 level or activity. To examine the functional connection between POLR2A and RBX1, we first validated their physical interaction in the cell. Western blot analysis identified POLR2A in the RBX1-associated immuno-precipitate of the DU145 cell lysates (Fig. 5a). Reciprocal analysis in which immunoprecipitated POLR2A-associated proteins were probed for RBX1 as well as other members of CRL complexes also confirmed the POLR2A-RBX1 interaction (Fig. 5a and Supplementary Fig. 4b). We hypothesized that POLR2A may be the ubiquitination substrate of RBX1-associated CRL complexes. Indeed, overexpression of RBX1 significantly increased the ubiquitination level of POLR2A, while silencing RBX1 had the opposite effect (Fig. 5b, c). There are seven lysine residues on ubiquitin as potential points of ubiquitination, in which K48-linked ubiquitination targets proteins for degradation, whereas K63-linked ubiquitination does not affect the stability of protein substrates, but often regulates their functions[35–37]. As we did not detect any significant changes of POLR2A protein levels upon RBX1 overexpression or downregulation (Supplementary Fig. 4c), we postulated that the RBX1-mediated ubiquitin modification on POLR2A may be non-K48-linked. Wildtype, K48R or K63R mutant form of His-tagged ubiquitin was transfected into DU145 cells with Dox-inducible expression of RBX1 shRNA. Ubiquiti-nated POLR2A was isolated by immobilized metal affinity chro-matography and detected by anti-POLR2A antibody. Ubiquitinated POLR2A levels were considerably decreased after Dox treatment in the cells expressing wildtype or K48R ubiquitin, but not in the cells expressing K63R ubiquitin, suggesting K63-linked ubiquitination is the primary form for RBX1-associated ubiquitination of POLR2A (Fig. 5d). Conversely, overexpression of RBX1 significantly increased ubiquitinated POLR2A levels in the presence of wildtype and K48R ubiquitin, but not K63R ubiquitin (Fig. 5e). To further verify the K63-associated ubiqui-tination activity of RBX1, we analyzed the ubiquitination of POLR2A labeled with wildtype, K48-only or K63-only ubiquitin (all other lysine residues are replaced with arginine residues except for K48 or K63). K63-only ubiquitin modification on

POLR2A was markedly reduced by 90% upon RBX1 depletion, in striking contrast to modest reduction of wildtype or K48-only ubiquitin modified POLR2A (Fig. 5f). Conversely, ectopic expression of RBX1 dramatically increased the level of K63-only ubiquitin-modified POLR2A, but not the level of K48-only ubi-quitin-modified POLR2A (Fig. 5g). Interestingly, depletion of the substrate recognition subunit SKP2, but not VHL, could partially phenocopy the effect of RBX1 knockdown on the K63-mediated ubiquitination of POLR2A, while overexpression of SKP2 pro-moted the K63-mediated ubiquitination of POLR2A (Supple-mentary Fig. 4d). This prompted us to think that SKP2 might be one of the substrate-recognizing subunits for the ubiquitination of POLR2A. To prove it, we performed in vitro pull-down experi-ments. We observed that purified GST-SKP2 was able to pull down a considerable amount of POLR2A compared to the well-documented POLR2A interaction protein POLR2H[38], indicating the direct interaction of SKP2 with POLR2A (Supplementary Fig. 4e). Moreover, by single plane confocal microscopy, we found both SKP2 and RBX1 show significantly positive colocalization with POLR2A (Supplementary Fig. 4f). Consistently, SKP2 knockdown partially mimicked the RBX1's knockdown effect on the 17p$^{loss}$ cell proliferation (Supplementary Fig. 4g). The weaker effect of SKP2, compared to that of RBX1, suggest that POLR2A may indeed interact with other CRL E3 ligases in addition to SKP2.

**RBX1 activates the POLR2A-dependent RNA transcription.** As the catalytic subunit of RNAP2, POLR2A is indispensable for mRNA synthesis. We asked whether RBX1 modulates the activity of RNAP2 via K63-linked ubiquitination of POLR2A. An isogenic pair of DU145 cells were incubated with 5-Ethynyl uridine (EU), a nucleotide analog for uridine, which is incorporated into nas-cent mRNA during active RNA synthesis. EU contains an alkyne that can react with an azide-containing fluorescent dye for monitoring mRNA synthesis temporally and spatially[39]. We found that RBX1 knockdown, indicated by red fluorescence protein (RFP) levels co-expressed with RBX1 shRNAs, impaired the global transcription in both 17p$^{loss}$ cells and 17p$^{neutral}$ cells (Fig. 6a and Supplementary Fig. 5a). As expected, the global transcription is associated with the levels of K63-linked, but not K48-linked ubiquitination of POLR2A (Fig. 6b and Supplemen-tary Fig. 5b). However, RBX1 depletion had more dramatic effect in the 17p$^{loss}$ cells with ~75% of suppression on total RNA synthesis, in comparison with ~25% of suppression in the 17p$^{neutral}$ cells (Fig. 6a). This is probably due to the abundant expression of POLR2A in the cell, which serves as a reservoir to maintain essential cell activities during proliferation. Specifically, in contrast to parental cells, RBX1 depletion in the 17$^{loss}$ cells resulted in much more inhibitory effects on the expression of those short-lived mRNAs[40] transcribed from genes such as *FOS*, *E2F3*, and *SNAI1*, compared to the control 5s rRNA levels (Fig. 6c and Supplementary Fig. 5c). Moreover, we used an in vitro RNAP2-dependent run off assay to measure the de novo tran-scription in vitro[41]. The HeLaScribe DNA template was incubated with nuclear extracts from the isogenic DU145 cells expressing Dox-inducible RBX1 shRNAs, and a transcription reaction without nucleoside triphosphates served as a negative control. The newly synthesized RNA was purified and quantified. Without Dox-induced RBX1 depletion, 17p$^{loss}$ cells had a relatively modest reduction on their mRNA transcriptional activity in comparison with the parental cells (Fig. 6d), suggesting that heterozygous 17p loss does not significantly impact cell activities, as observed in many cancer cells with this genomic event. However, further depletion of RBX1 in the 17$^{loss}$ cells resulted in much more severe effects on the mRNA transcription (Fig. 6d).

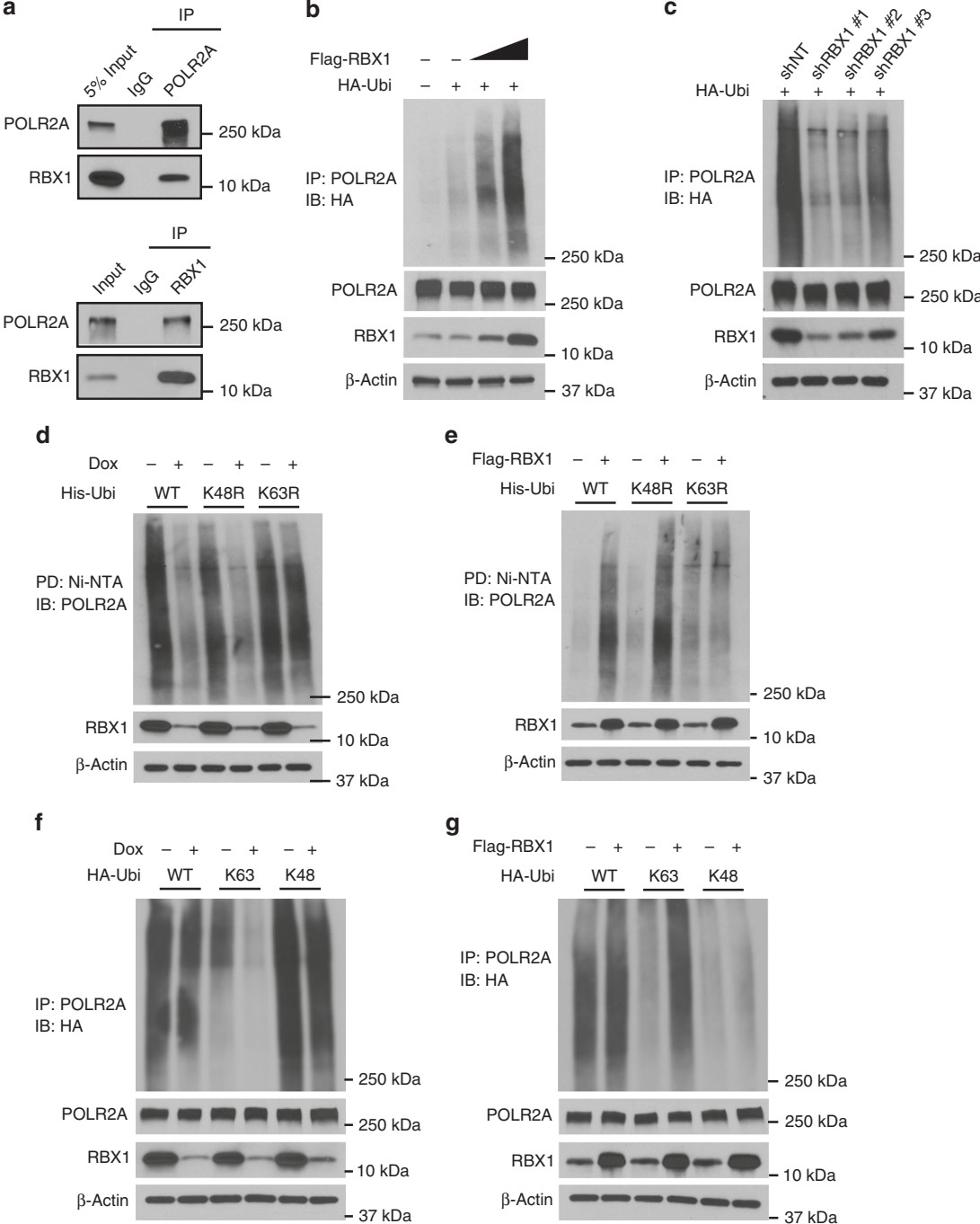

**Fig. 5** RBX1 modifies POLR2A by the K63-linked ubiquitination. **a** RBX1 physically interacts with POLR2A. Immunoprecipitation (IP) and western blot analyses were performed using indicated antibodies. Normal IgG was used as a negative control for IP. **b**, **c** RBX1 overexpression (**b**) or knockdown (**c**) increases or decreases the level of ubiquitinated POLR2A, respectively. DU145 cells transfected with indicated expression vectors were treated with MG132 and ubiquitinated POLR2A was pulled down (IP) and subject to immunoblotting (IB) analysis. **d**, **e** RBX1-mediated ubiquitination acts on the lysine-63 (K63) residue of POLR2A. DU145 cells were transfected with the indicated ubiquitin expression vectors and treated with MG132. Cells were also treated with Dox to induce RBX1 knockdown in **d**, or ectopic Flag-RBX1 was expressed in the cells in **e**. Ubiquitinated POLR2A was pulled down (PD) with Ni-NTA Agarose and subject to immunoblotting (IB) analysis. **f**, **g** RBX1 knockdown (**f**) or overexpression (**g**) decreases or increases the level of lysine-63 (K63) ubiquitination of POLR2A, respectively. Equal amounts of cell lysates were analyzed by immunoprecipitation and IB assays as described above. Data are representative of three independent experiments

**RBX1 depletion-sensitized 17p^loss CRPC to POLR2A inhibition.** Our recent studies have shown that collateral deletion of *POLR2A* with *TP53* in human colorectal cancer renders them susceptible to further inhibition of POLR2A[22]. Similarly, as observed in colorectal cancer, the *POLR2A* gene is almost always co-deleted with *TP53* in the 17p deletion region in prostate cancer, and the expression levels of *POLR2A*, but not *TP53*, are significantly correlated with the loss of 17p (Supplementary Fig. 6a, b). In addition, this tight correlation between *POLR2A* copy numbers and protein levels were further validated in a panel

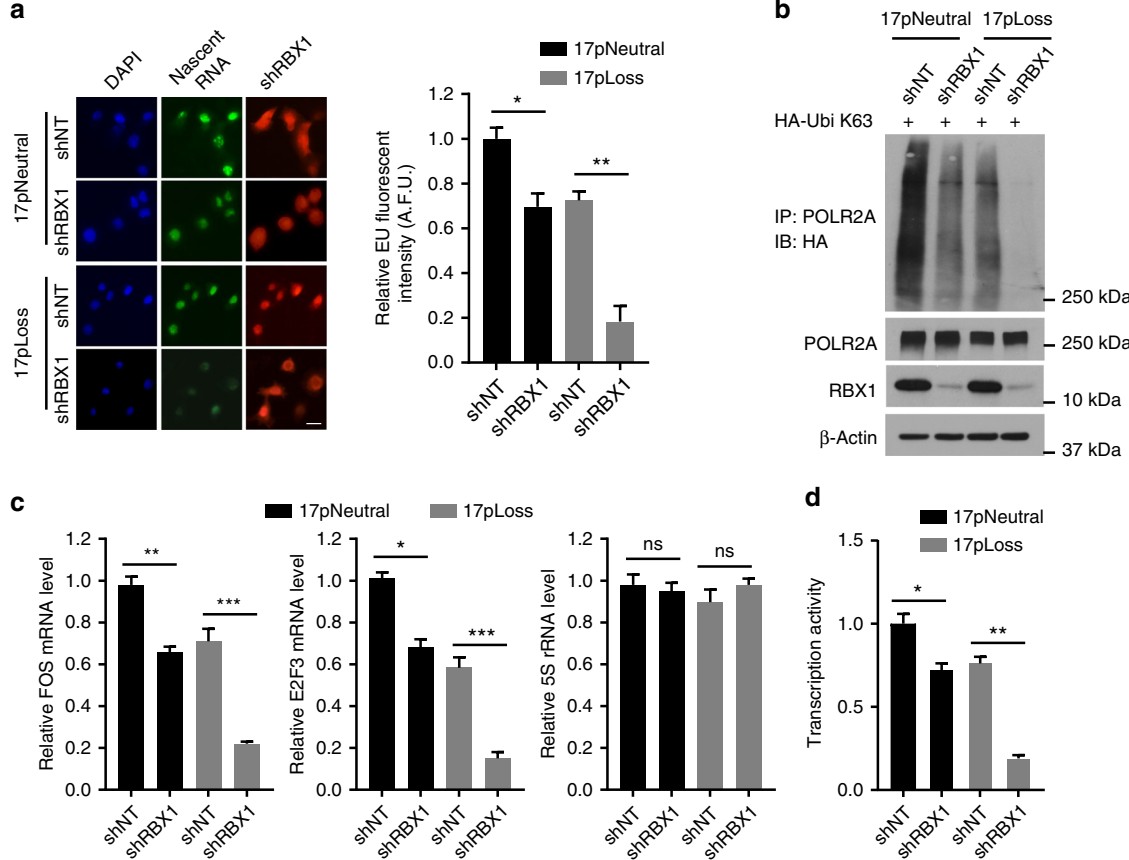

**Fig. 6** RBX1 promotes the POLR2A-dependent RNA transcription. **a** Global RNA synthesis in the parental and isogenic 17p$^{loss}$ DU145 cells was evaluated by measurement of 5-EU incorporation after Dox-inducible RBX1 knockdown and the intensity was quantified with the CellProfiler Software. Scale bar, 25 μm. **b** Effect of RBX1 knockdown on the levels of lysine-63 (K63) ubiquitination of POLR2A in the parental and isogenic 17p$^{loss}$ DU145 cells. **c** qPCR analysis of the levels of short-lived mRNA (FOS and E2F3) compared to the control 5S rRNA in the isogenic DU145 cells with or without Dox-inducible RBX1 knockdown. **d** qPCR analysis of the in vitro transcription activity using HelaScribe DNA template. Data are representative of three independent experiments and analyze by unpaired two-tailed $t$-test. Error bars denote SD. ns not significant; **, $p < 0.01$; ***, $p < 0.001$

of human prostate cancer cell lines as well as a human tumor tissue microarray including 169 prostate tumor samples (Fig. 7a and Supplementary Fig. 6c, d). In comparison with the 17p$^{neutral}$ cells (22Rv1, DU145), treatment of α-amanitin, a specific inhibitor of POLR2A, had markedly higher levels of cell-killing effect on the 17p$^{loss}$ cells (PC-3, VCaP). The half-maximum inhibitory concentration (IC50) was ~0.3 μg ml$^{-1}$ for the 17p$^{loss}$ cells, which was 5–10 fold lower than that of the 17p$^{neutral}$ cells (Fig. 7b and Supplementary Table 1). By contrast, 17p$^{loss}$ cells did not show any greater sensitivity to the treatment of actinomycin D, a nonspecific transcription inhibitor[42] (Fig. 7c). The results suggest that inhibiting POLR2A by α-amanitin specifically kills the prostate cancer cells harboring 17p loss.

As the 17p$^{loss}$ cancer cells are vulnerable to the inhibition of RBX1, we asked whether RBX1 inhibition can further improve the efficacy of α-amanitin-based drug through combined action in treating CRPC with 17p loss. To test it, the isogenic pair of 17p$^{neutral}$ and 17p$^{loss}$ DU145 cell lines expressing Dox-inducible RBX1 shRNAs were treated simultaneously with Dox and α-amanitin at a variety of combined doses (Fig. 7d, e). Dox-induced inhibition of RBX1 exerted significantly enhanced cytotoxicity in the 17p$^{loss}$ DU145 cells, but only had relatively modest effect on their parental DU145 cells without 17p loss. A combination of 0.1 μg ml$^{-1}$ of α-amanitin and Dox caused massive cell death (over 80%) in the 17p$^{loss}$ cells. By contrast, the same combined treatment only had a modest effect on the survival of the

17p$^{neutral}$ cells. As a control, α-amanitin alone only killed 50% of 17p$^{loss}$ cells at the concentration of 1.0 μg ml$^{-1}$ (Fig. 7d, e). Similarly, the 17p$^{loss}$ cells (PC3 and VCaP), upon Dox-induced depletion of RBX1, were more sensitive to α-amanitin treatment, showing a synergistic effect with a combination index (CI) <0.5 (Fig. 7f). The results clearly demonstrated that inhibition of RBX1 markedly reduced the effective dose of α-amanitin and achieved better cell killing effect in the 17p$^{loss}$ cancer cells.

**RBX1 inhibition potentiates the efficacy of α-amanitin ADC.** Based on the synergy of RBX1 depletion and α-amanitin-mediated inhibition, we investigated the efficacy of the combined treatment in 17p$^{loss}$ CRPC in vivo. The 17p$^{loss}$ DU145 cells expressing Dox-inducible RBX1 were orthotopically implanted into the prostate of NOD/SCID mice. Once tumors had been established for 10 days post implantation, the tumor-bearing mice were randomized and administrated with Dox food and α-amanitin-conjugated anti-EpCAM antibodies[43] (ADC, weekly, twice in total) (Fig. 8a and Supplementary Fig. 7). Similar to our previous data from colorectal cancer models[22], the treatment of ADC alone inhibited ~84% of tumor growth at the dose of 10 μg kg$^{-1}$ (corresponding to the amount of α-amanitin in the ADC). Dox-induced depletion of RBX1 alone also exerted notable inhibition on the tumor growth. However, combinatorial treatment of ADC (3.0 or 1.0 μg kg$^{-1}$) and Dox profoundly intensified

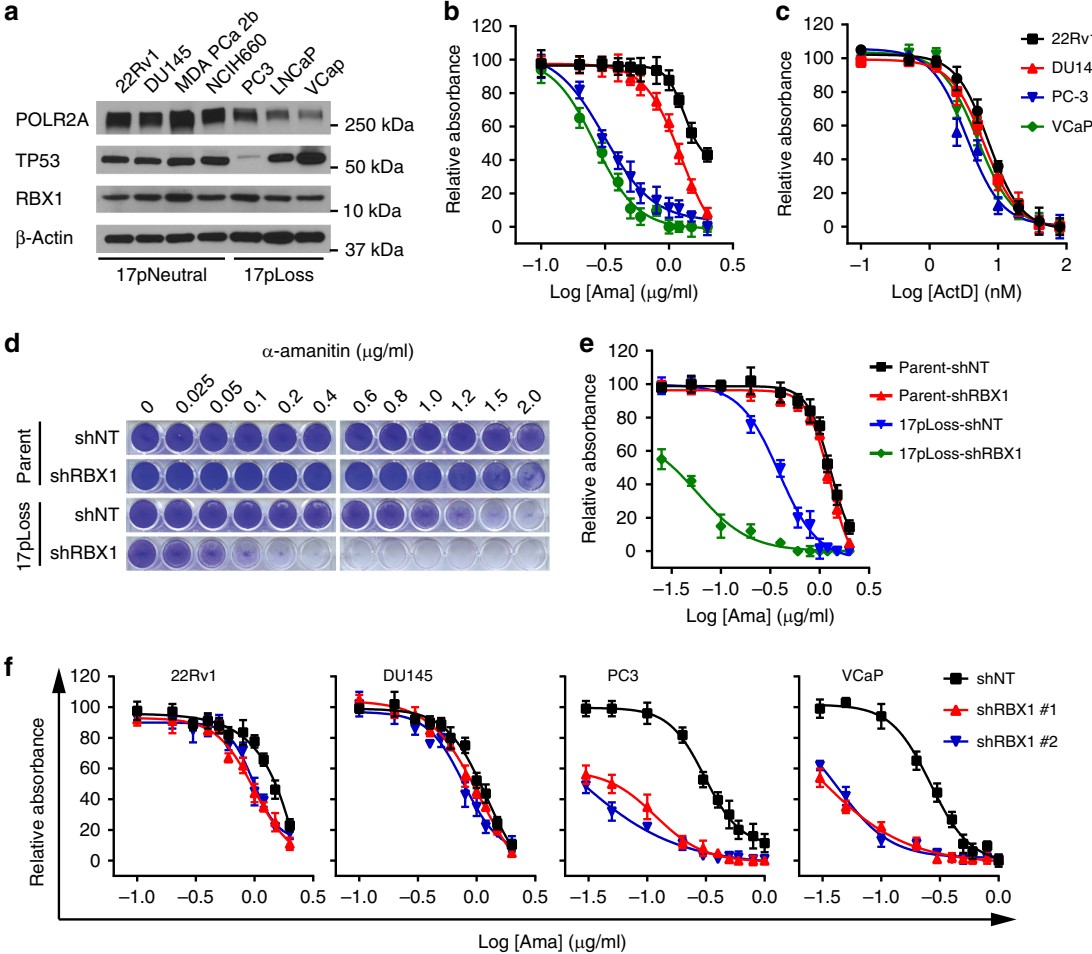

**Fig. 7** RBX1 depletion sensitized 17p$^{loss}$ prostate cancer cells to POLR2A inhibition. **a** Protein levels of POLR2A, p53, Rbx1, and β-Actin in human prostate cancer cell lines. **b**, **c** Cell proliferation of 17p$^{neutral}$ (22Rv1 and DU145) and 17p$^{loss}$ cells (PC3 and VCaP) treated with α-amanitin (**b**) or actinomycin D (**c**). **d**, **e** RBX1 depletion sensitizes the 17p$^{loss}$ DU145 cells to the treatment of the POLR2A inhibitor, α-amanitin. Representative images (**d**) and quantitative results (**e**) of cell survival are shown. **f** Cell proliferation of human prostate cancer cell lines under α-amanitin treatment. 17p$^{neutral}$ (22Rv1 and DU145) and 17p$^{loss}$ (PC3 and VCaP) cells, with or without Dox-induced RBX1 knockdown, were treated with increasing doses of α-amanitin. Data are representative of three independent experiments

the suppression of tumor growth, leading to nearly complete (>90% reduction) tumor regression (8/8 or 6/8 mice, respectively). Even combined with a low dose of ADC (0.3 μg kg$^{-1}$), RBX1 depletion had comparable effects to that of the treatment with 3.0 μg kg$^{-1}$ of ADC alone (Fig. 8b, c). IHC analyses confirmed that the combined treatment inhibited the tumor cell proliferation and promoted their apoptosis, to a much greater extent than the single agent ADC or Dox (Fig. 8d and Supplementary Fig. 8a, b). The combinatorial treatment had no notable toxicity in vivo as reflected by negligible changes of body weights or blood liver enzymes (Supplementary Fig. 8c, d). Collectively, these results suggested that heterozygous loss of 17p confers therapeutic vulnerability of human CRPC to the inhibition of POLR2A (on the 17p) and RBX1 (not on the 17P), and that inhibiting RBX1 significantly sensitizes the CRPC to the treatment of α-amanitin-antibody conjugates.

## Discussion
The common treatment option for prostate cancer is to deprive the levels of androgen by surgical castration or androgen-deprivation therapy[2,4–6]. Hormone therapy is usually continued in patients with metastatic prostate cancer. However, nearly all

the patients with metastatic prostate cancer eventually develop CRPC. Therapeutic options for CRPC are often limited to chemotherapy with the drugs such as docetaxel and cabazitaxel[2,4–6]. The large-scale and multi-dimensional analyses of human prostate cancer genomics now provide comprehensive profiles of the cancer genomic alterations, which enables the development of therapies that target these changes as well as prognosis that identifies patients who may benefit from these therapies[13]. Identifying tumor vulnerabilities of CRPC would provide novel therapeutic approaches for this incurable disease. Similar to other types of cancer, the tumor suppressor *TP53* gene is also frequently inactivated by mutation or deletion in human prostate cancer. One of the major efforts in cancer therapeutics is to restore p53 activity in cancer cells. Different strategies have been taken to develop small molecule compounds that specifically target mutant p53 or p53 inhibitors such as Mdm2 and Mdmx[44,45]. However, no effective p53-based therapy has been successfully translated from bench to bedside due to the complexity of p53 signaling. Therefore, identification of vulnerabilities conferred by *TP53* deletion or mutation is a major challenge to target p53 aberrancy in human cancers including CRPC.

Our recent study demonstrated that genomic deletion of *TP53* frequently encompasses a neighboring essential gene, *POLR2A*,

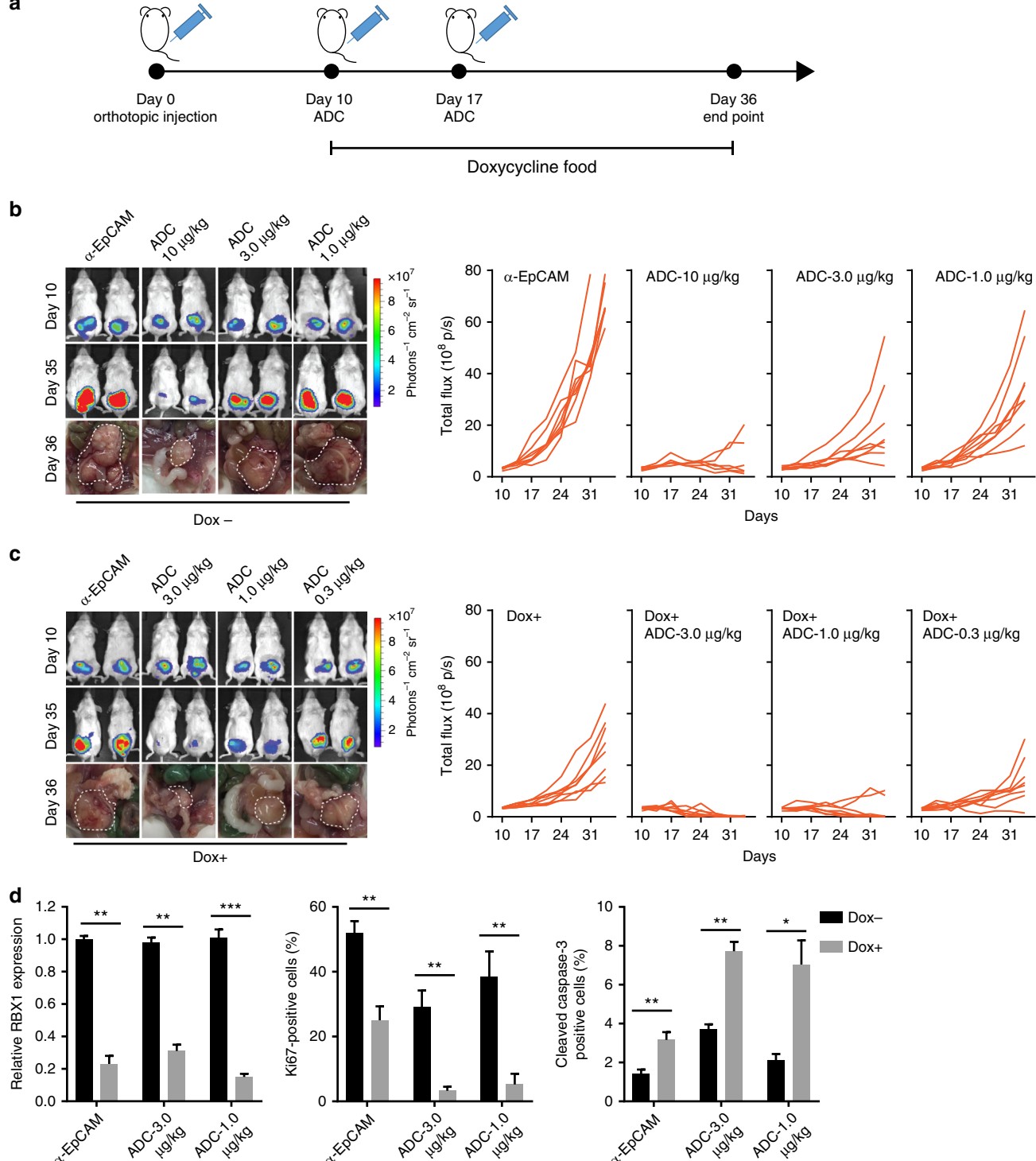

**Fig. 8** Inhibition of RBX1 sensitizes 17p$^{loss}$ CRPC to the treatment of α-amanitin-based ADC. **a** Schematic illustration of orthotopic injection of 17p$^{loss}$ DU145 cells (1 × 10$^6$ cells) followed by twice i.p. injection of ADC and Dox food treatment. **b**, **c** Representative bioluminescent tumor images and individual tumor growth curves of xenograft tumors derived from orthotopically implanted 17p$^{loss}$ DU145 cells without (**b**) or with (**c**) Dox treatment. Once tumor was established, mice were randomly divided to 4 groups (n = 8) and treated with either free anti-EpCAM antibody or different doses of anti-EpCAM-amanitin conjugates (ADC). **d** Quantification of RBX1 knockdown efficiency, cell proliferation (Ki-67 staining) and apoptosis (cleaved caspase-3 staining) in the xenografted tumor tissues described above. Data are analyze by unpaired two-tailed t-test and are presented as the mean ± SD. **, p < 0.01; ***, p < 0.001

rendering cancer cells with heterozygous *TP53* deletion vulnerable to further suppression of POLR2A[22]. Of note, the *TP53*-containing deletion often span over 20 megabases on the 17p and more than 200 genes are co-deleted, which may impact prostate

tumor progression and metastasis. While it is impractical to delineate individual gene contribution and their interactions in tumorigenesis, we reasoned that the 17p loss may create therapeutic vulnerabilities as some of the genes are in charge of

essential cellular activities, as observed for *POLR2A*. A number of studies have shown that CRISPR-based screens are a systematic approach to identify essential genes required for cell survival and proliferation[46–48]. Although a majority of the essential genes are shared between different cancer cell types, some of them are essential only in the context of specific genomic profiles, suggesting that essentiality is not an intrinsic property of a gene but is rather impacted by genetic and environmental factors. As an example, analysis of differentially essential genes between Ras-dependent and -independent cell lines uncovered PREX1 as an acute myeloid leukemia-specific activator of the oncogenic Ras-MAPK signaling[49]. On this basis, we proposed that prostate cancer cells with 17p loss would have their specific gene dependency compared to their counterparts without 17p loss. In this study, *RBX1* stands out as an essential gene associated with 17p loss in CRPC cells.

As a key catalytic subunit of RNAP2, POLR2A contains a carboxyl terminal domain (CTD) composed of 52 heptapeptide repeats (YSPTSPS) that are essential for the RNA polymerase activity[50]. Phosphorylation of the heptapeptide repeats regulates the activity of POLR2A. The phosphorylation state is believed to result from the balanced action of CTD kinases (CDK7 and CDK9) and the Ser/Thr phosphatase CTDSP1. We found that the E3 ubiquitin ligase RBX1 is a new direct interactor of POLR2A. RBX1 promotes the enzymatic activity of POLR2A via K63-linked poly-ubiquitination. In support of its functional essentiality, RBX1 expression levels are inversely correlated with POLR2A in human prostate cancers ($r = -0.45$, TCGA), suggesting that loss of POLR2A in the 17p$^{loss}$ cells is probably compensated by the upregulated expression of RBX1 in prostate tumors. Inhibition of RBX1 markedly enhanced the effectivity of α-amanitin-conjugated ADC in treating CRPC in vivo, indicating the translational potential of the RBX1-targeted therapeutic approach.

Results of CRISPR-based knockout (Fig. 2) or silencing of RBX1 (Figs. 3 and 4) in CRPC cell lines revealed that RBX1 is essential for proliferation only in 17p$^{loss}$ cells, but not in 17p$^{neutral}$ cells. These findings suggest that RBX1 is not an essential gene for proliferation at least in DU145 cells. However, RBX1 was previously shown to be an essential gene required for mouse embryonic development[51]. In addition, CRISPR-based genome-wide analysis showed that RBX1 is an essential gene for those four cell lines tested (KBM7, K562, Jiyoye, and Raji)[52,53]. To understand this discrepancy, we have analyzed RBX1 genetic integrity in a large set of tumor cells using Cancer Cell Line Encyclopedia (CCLE) database. The results revealed homozygous deletion of RBX1 in at least 11 cell lines, including ACHN, HT115, KALS-1, LCLC-103H, MSTO-211H, REC-1, SH-SY5Y, SK-CO-1, SK-N-SH, SNU-719, and TE-1. Moreover, KBM7 is a haploid cell line that only contains one copy of Chr17, while K562 is a cell line with heterozygous loss of 17p. Of note, the genomic data of Jiyoye is unavailable from CCLE, whereas Raji cells seem to contain neural copy number of 17p. Collectively these data suggest that the essentiality of RBX1 for cell growth may be context-dependent. Given our work on DU145 and the CCLE analysis of 11 tumor cell lines, it appears that RBX1 is not essential for global cell survival in a significant set of tumor cell lines. Moreover, our study has raised possibility that hemizygosity may be a significant factor that sensitizes RBX1's essentiality. Results from CRISPR experiments with the haploid KBM7 and the17p-deficient K562 cells from the Wang study appear in line with our assertion. Finally, both RBX1 and its paralog RBX2 (also named as RNF7), which shares 7 out of 8 Cys/His residues that constitute their RING finger domain, are found to be evolutionarily conserved among many species from yeast to humans[54]. More importantly, either human RBX1 or RBX2 can rescue the lethal phenotype caused by deletion of the RBX ortholog Hrt1 in yeast, indicating they are functionally redundant to some extent[29].

Beside the recent approval of Kadcyla (T-DM1) and Adcetris (SGN-35), a number of ADCs have entered clinical trials[55–57]. However, most ADCs are based on a few toxic compounds, such as auristatins and maytansines, and their mechanisms of action are often limited to the microtubule disruption. The clinical efficacy of these ADCs could suffer from limited activity in different cancer indications and in resistant cancer cells. Accordingly, the use of new drugs that function via alternative toxicity mechanisms will potentially enhance the therapeutic activity of ADCs. As one of the newly developed warheads in ADCs, α-amanitin has advantages over other commonly used toxins because of its unique mode of action and molecular characteristics[22,43]. In particular, as the most potent inhibitor of cellular transcription, α-amanitin targets the unique vulnerability of cancer cells conferred by genomic loss of POLR2A and 17p, which significantly increases therapeutic index by selectively killing cancer cells with this genomic aberrancy, and accordingly reduces potential in vivo toxicity. Inhibition of RNAP2 by α-amanitin leads not only to apoptosis of dividing cells, but also to that of slowly growing cells, as often observed in metastatic prostate cancer. Therapeutic strategy using RBX1 inhibitors in combination with the α-amanitin-conjugated ADCs will integrate cancer genomics into precision medicine to fuel exciting progress against a challenging class of human cancers. Heterozygous loss of *TP53* and 17p is also found in other types of cancer, suggesting that α-amanitin-based ADCs may be great drug candidates in those cancers. In the past decade, the ubiquitin–proteasome system has emerged as an attractive target for the development of novel therapeutics[58,59]. E3 ubiquitin ligases are particularly valuable targets because they confer substrate specificity on the ubiquitin system. RBX1 is a key component that binds and ubiquitinates substrate proteins in CRL complexes. Further structural investigation of the RBX1-containing CRLs will allow us to better understand the assembly and structure of CRL complexes, and empower the design of small-molecule inhibitors and modulators of CRL activity.

## Methods

**Tissue culture and tissue microarray**. HEK293T, 22Rv1, DU145, MDA PCa 2b, NCI H660, PC3, LNCaP, VCaP, MEG01, KU812, and K562 cell lines were purchased from the American Type Culture Collection (ATCC). KBM7 cell lines was ordered from Horizon Discovery. All cell lines were maintained under standard conditions specified by the manufacturer and were tested negative for mycoplasma contamination using the Mycoplasma detection kit (Lonza). Prostate tumor tissue microarray (PR8011bt, PR1921a, PR632) were purchased from Biomax, including 228 prostate tumor samples and 51 normal prostate tissue samples.

**Antibodies**. Anti-RBX1 antibody (ab133565, 1:1000 dilution for immunoblotting and 1:100 for IHC) was purchased from Abcam. Anti-POLR2A antibody (sc-47701,1:5000 dilution for immunoblotting and 1:200 for IHC), anti-CUL1 antibody (sc-17775, 1:1000 dilution), anti-CUL2 antibody (sc-166506, 1:1000 dilution), anti-CUL4A/4B antibody (sc-377188, 1:1000 dilution), anti-CUL5 antibody (sc-373822, 1:1000 dilution) and anti-CUL7 (sc-53810, 1:1000 dilution) antibody were obtained from Santa Cruz. Anti-Ki67 antibody (#12075, 1:100 dilution), anti-cleaved Caspase-3 (Asp175) (#9664, 1:1000 dilution), Anti-SKP2 antibody (#2652, 1:1000 dilution), Anti-VHL antibody (#68547, 1:1000 dilution) and Anti-CUL3 (#2759, 1:1000 dilution) were purchased from Cell Signaling Technology. Anti-p53 (sc-126, 1:1000 dilution), anti-actin (sc-1616, 1:5000 dilution), HRP-anti-mouse IgG (sc-2055, 1:5000 dilution), HRP-mouse IgG kappa binding protein (sc-516102, 1:5000 dilution) and HRP-anti-rabbit IgG (sc-2054, 1:5000 dilution) were purchased from Santa Cruz.

**shRNA interference**. Lentiviral pGIPZ vectors expressing non-silencing shRNA control, shRBX1 and shGTF2H1 were purchased from Dharmacon (originally from Open Biosystems). The hairpin sequence in the pGIPZ shRBX1 #1 were cloned into the pTRIPZ (Dharmacon) using standard protocol provided by the manufacturer. The TRIPZ vector is a Dox-inducible system with a red fluorescent protein reporter. The shRNA clone identification numbers and shRNA sequences are as below:

shRBX1 #1 (V3LHS_637677, 5′-GCATTAAAGCAGCGTATC-3′)
shRBX1 #2 (V3LHS_405194, 5′-GCATTAAAGCAGCGTATC-3′)

shRBX1 #3 (V3LHS_337320, 5′-GCATTAAAGCAGCGTATC-3′)
shGTF2H1 #1 (V3LHS_399134, 5′-GCATTAAAGCAGCGTATC-3′)
shGTF2H1 #2 (V3LHS_399137, 5′-GCATTAAAGCAGCGTATC-3′)
shSKP2 #1 (V2LHS_199552, 5′-TATCACTTAAGTCTAGATG-3′)
shSKP2 #2 (V2LHS_12473, 5′-ATACTTCATAGACAACTGG-3′)
shVHL #1(V2LHS_202399, 5′-TAATGAATCTAAGTCTAGC-3′)
shVHL #2(V2LHS_262050, 5′-AATTCTCAGGCTTGACTAG-3′)

**Generation of sgRNA-expressing vectors.** Briefly, the vector DNA expressing Cas9 and sgRNA was digested with BbsI and treated with alkaline phosphatase. The linearized vector was then gel purified. Meanwhile, the pair of oligo DNA for sgRNA targeting POLR2A was annealed and phosphorylated. The sequences of oligo DNA are as follows: 5′-CAGCCGACTGAACAGCCGTA-3′ (WDR81) and 5′-GAGGCAGACTCACGTGGGGT-3′ (MAP2K3). The annealed sgRNA was subsequently ligated to the linearized vector. Genome-editing efficacy was tested by Suveyor assay[60].

**Immunoblotting.** Cell extracts were prepared using cell lysis buffer (50 mM Tris, pH 7.5, 150 mM NaCl, 1 mM EDTA, 0.5% NP-40, 0.5% Triton X-100, 1 mM phenylmethylsulfonyl fluoride, 1 mM sodium fluoride, 5 mM sodium vanadate, 1 µg ml$^{-1}$ of aprotinin, leupeptin, and pepstatin). Proteins were resolved by SDS-polyacrylamide gel electrophoresis gels and then proteins were transferred (Bio-Rad) to polyvinylidene difluoride membranes (Millipore). After blocked with 5% milk, membranes were incubated with indicated primary antibodies (1:1000 dilution). Subsequently, membranes were washed and incubated with peroxidase-conjugated secondary antibodies (Santa Cruz Biotechnology). Finally, the relevant protein was visualized by enhanced chemiluminescence system (PerkinElmer) according to the manufacturer's instructions. Uncropped scans of the western blots are presented in the Supplementary Fig. 9.

**Immunoprecipitation.** Cells were lysed on ice for 30 min in immunoprecipitation buffer (1% NP-40, 50 mM Tris-HCl, 500 mM NaCl and 5 mM EDTA) containing protease inhibitor cocktail. Protein concentration was determined by the BCA assay (Thermos). Cell lysates (800 µg) were incubated with 2 µg of indicated antibodies or control normal IgG at 4 °C overnight with rotary agitation. Protein A/G-sepharose beads were then added to the lysates and incubated for another 4 h. Beads were washed three times with ice-cold immunoprecipitation buffer and boiled for 10 min in 1× sample loading buffer. Total cell lysates and immuno-precipitates were resolved by SDS-polyacrylamide gel electrophoresis and analyzed either by western blotting or stained with silver nitrate. Protein bands were excised and subjected to mass spectrometry analysis at the MD Anderson Proteomics and Metabolomics Core.

**GST pull-down assay.** Two micrograms of GST-POLR2H (LSBio, G29812), GST-SKP2 (LSBio, G30419), or control GST protein (Sigma, SRP5348) was immobilized on agarose beads (Glutathione Sepharose 4B) and incubated with 200 µg of lysates from DU145 cells at 4 °C for 2 h. Bound protein was washed five times with NETN buffer (1% NP-40, 50 mM Tris-HCl, 500 mM NaCl and 5 mM EDTA) and was subjected to SDS-PAGE gel electrophoresis and detected by using anti-POLR2A antibody.

**Immunohistochemistry.** Briefly, samples were deparaffinized and rehydrated followed by antigen retrieving used 0.01 M sodium-citrate buffer (pH 6.0) at a sub-boiling temperature for 10 min. The sections were then incubated with 3% hydrogen peroxide for 10 min to block endogenous peroxidase activity. After 1 h of pre-incubation in 5% normal goat serum, the samples were incubated with indicated primary antibodies at 4 °C overnight. The sections were then washed three times and incubated with secondary antibodies. Counterstaining color was carried out using Harris modified haematoxylin. All immunostained slides were scanned on the Automate Cellular Image System III for quantification by digital image analysis.

**Immunofluorescence.** Cells grown on coverslips were fixed in 3% paraformaldehyde in phosphate-buffered saline (PBS) at room temperature for 20 min. After fixation, cells were subjected to 0.5% Triton X-100 for 10 min and then blocked with 5% goat serum in PBS for 1 h at room temperature. Coverslips were immunostained with indicated primary antibodies in 5% goat serum in wet chamber at 4 °C overnight. Coverslips were then washed and incubated with secondary antibodies (Alexa Fluor 594 and Alexa fluor 488) for 1 h at room temperature. Cells were then stained with DAPI to visualize nuclear DNA. The coverslips were mounted onto glass slides with anti-fade solution. Single plane confocal images were acquired with Olympus FV-1000 MPE system (Center Valley, PA). Colocalization analysis was performed with the use of IMARIS software colocalization module (Bitplane, Concord, MA).

**Genomic DNA isolation and copy number validation.** Total genomic DNA was extracted either from cell lines using DNeasy Blood & Tissue Kit (Qiagen) or from human tissue specimen using QIAamp DNA FFPE Tissue Kit (Qiagen) according to the manufacturer's instructions. The copy number variations for *PSMB6, TP53, POLR2A*, and *FLCN* were determined using TaqMan probes and TaqMan PCR kit. The reference gene *TERT* was quantified in the same reaction at the same time for each DNA sample.

**RNA isolation and quantitative PCR.** Total RNA were isolated using RNeasy Micro kit (Qiagen) and then reverse-transcribed using iScript cDNA Synthesis Kit (Bio-Rad). The resulting cDNA was used for qPCR using iTaq Universal SYBR Green Supermix (Bio-Rad) with specific primers for each gene and the results were normalized to 18 S rRNA control. The primer sequences are:
RBX1: 5′-TTGTGGTTGATAACTGTGCCAT-3′; 5′-GACGCCTGGTTAGCT TGACAT-3′; GTF2H1: 5′-GACCTTGTTGTGAGTCAAGTGA-3′; 5′-CCTGCTT ATGATTGGATGTGGAA-3′; FOS: 5′-CCGGGGATAGCCTCTCTTACT-3′; 5′-C CAGGTCCGTGCAGAAGTC-3′;
E2F3: 5′-AGAAAGCGGTCATCAGTACCT-3′; 5′-TGGACTTCGTAGTGCAG CTCT-3′;
SNAIL1: 5′-TCGGAAGCCTAACTACAGCGA-3′; 5′-AGATGAGCATTGGC AGCGAG-3′;
5S rRNA: 5′-GGCCATACCACCCTGAACGC-3′; 5′-CAGCACCCGGTATTC CCAGG-3′;
18S rRNA: 5′-TGTGCCGCTAGAGGTGAAATT-3′; 5′-TGGCAAATGCTTTC GCTTT-3′.

**Cell survival assay.** Equal numbers of cells were plated in 12-well plates in triplicate. After 24 h cells were treated with α-amanitin or actinomycin D at indicated concentrations for another 72 h. Cells were then fixed with 10% methanol and stained with 0.1% crystal violet (dissolved in 10% methanol). After staining, wells were washed three times and destained with acetic acid. The absorbance of the crystal violet solution was measured at 590 nm.

**Focused CRISPR-Cas9 genetic screening.** The CRISPR-Cas9 screen was performed using the Human CRISPR enriched pooled library for nuclear proteins (Addgene # 51047), which contains 37,330 guides and an average of 10 guides per gene[46,47,61]. Briefly, isogenic parental or 17p$^{loss}$ DU145 cells were transduced with Dox-inducible FLAG-Cas9 vector. A single colony with robust Cas9 expression (>50-fold induction) upon doxycycline treatment was selected for genetic screening. A total of 90 million DU145 cells were transduced with viral supernatant at MOI of 0.3 and selected with blasticidin for 24 h after infection for 3 days. Ten million cells were harvested for genomic DNA extraction 24 h after infection. The remaining cells were cultured in medium containing 1 µg ml$^{-1}$ doxycycline for 14 doublings before being harvested for genomic DNA extraction. Nested PCR was performed on genomic DNA to extract sgRNAs information and subjected to HiSeq 2500 (Illumina) with a single-end 50 bp run. Data were analyzed by edgeR. The following nested PCR primers are used:
Outer Primer 1: 5′-GCCGGCTCGAGTGTACAAAA-3′
Outer Primer 2: 5′-AGCGCTAGCTAATGCCAACTT-3′
Inner Primer 1:
5′-CAAGCAGAAGACGGCATACGAGATCXXXXXTTTCTTGGGTAGT TTGCAGTTTT-3′
(XXXXX denotes the sample barcode)
Inner Primer 2:
5′-AATGATACGGCGACCACCGAGATCTACACCGACTCGGTGCCACT TTT-3′
Illumina Sequencing primer:
5′-CGGTGCCACTTTTTCAAGTTGATAACGGACTAGCCTTATTTTAACTTGC TATTTCTAG
-CTCTAAAAC-3′
Indexing primer:
5′-TTTCAAGTTACGGTAAGCATATGATAGTCCATTTTAAAACATAA TTTTAAAACTGCAAA
-CTACCCAAGAAA-3′.

**Differential representation analysis.** The bioinformatics analysis was modified from a recent report[26]. Briefly, counts for each sgRNA was first normalized using the following method for visualization:

$$d = \log2\left(\frac{count \times 10^6}{total\ reads} + 1\right)$$

The representation analysis was then conducted with Bioconductor edgeR package (v3.18.1) in statistical programming environment R (v3.4.1), and the exact statistical method was carried out to detect over/under-represented individual sgRNAs between 17p$^{loss}$ and 17p$^{neutral}$ groups, and FDR < 0.05 was considered as statistically significant, while FDR > 0.9 was considered no differences between two conditions (day 0 and day 14). Subsequently, the camera gene-set test was used to prioritize genes based on ranking of sgRNAs that target the same gene.

**Apoptosis analysis**. 22Rv1, DU145, PC3, and VCaP cell lines were treated with 1 µg ml$^{-1}$ doxycycline for 4 days and stained with annexin V-PE and 7-AAD (Biovision). Apoptosis was then analyzed by flow cytometry according to the manufacturer's instructions. Live cells (annexin V-negative, 7-AAD-negative), pre-apoptotic (annexin V-positive, 7-AAD-negative) and apoptotic cells (annexin V-positive, 7-AAD-positive) were included in the analysis.

**Amnis ImageStreamX**. The internalization of HEA125 (EpCAM) and Ama-HEA125 was assessed using imagestream flow cytometry. Briefly, $2.5 \times 10^6$ WI-38 and DU145 (parental or 17p$^{loss}$) cells were incubated with indicated antibody or ADC for 30 min at 4 °C in the dark. After washing twice with 1% BSA in PBS containing 0.02% Azide, the cells were stained with the secondary anti-Human IgG antibody conjugated with alexa fluor 594 (Ref A11014, Life technologies) for 20 min at 4 °C in the dark. The cells were washed twice and kept at 4 °C or placed in culture medium at 37 °C for 1 h. The cultured cells were washed again and all the samples were resuspended and analyzed immediately on the AMNIS Imagestream (Imagestream X Mark II Imaging cytometer, EMD Millipore). We acquired 10,000 cells for each condition. The cellular uptake was quantified using the Internalization wizard of the IDEAS software.

**Prostate tumor xenograft mouse model**. Male NOD/SCID mice (4–6 weeks old) were purchased from Jackson Laboratories. All studies were approved and supervised by the Institutional Animal Care and Use Committee of MD Anderson Cancer Center and Indiana University. Dox-inducible parental or isogenic 17p$^{loss}$ DU145 cancer cells ($1 \times 10^6$) in 100 µl PBS with 50% Matrigel (BD Biosciences) were injected subcutaneously into the dorsal flank using a 100-µl Hamilton microliter syringe. Once the tumors reached a palpable stage (~100 mm$^3$), the animals were randomized and treated with or without doxycycline food (200 mg kg$^{-1}$, Bio-Serv). Growth in tumor volume was recorded using digital calipers and tumor volumes were estimated using the formula $0.5 \times L \times W^2$, where $L$ is the longest diameter and $W$ is the shortest diameter. For orthotopic prostate tumor model, the NOD/SCID mice were anaesthetized with 2% isoflurane (inhalation) and 22Rv1 and PC3 cells ($1 \times 10^6$) expressing luciferase were implanted into the dorsal prostate[62]. Ten days after cell injection, mice bearing tumors were randomly divided into four groups and treated with or without doxycycline food as mentioned above. To determine whether RBX1 knockdown sensitizes prostate tumor cells to the treatment of ADC. Isogenic 17p$^{loss}$ DU145 cells ($1 \times 10^6$) cells stably expressing Dox-inducible RBX1 shRNA were implanted into the dorsal prostate of the male NOD/SCID mice. Ten days after cell injection, mice bearing tumors were randomly divided into eight treatment groups: (1) free anti-EpCAM antibodies; (2) 10.0 µg kg$^{-1}$ ADC; (3) 3.0 µg kg$^{-1}$ ADC; (4) 1.0 µg kg$^{-1}$ ADC; (5) Dox + free anti-EpCAM antibodies; (6) Dox + 3.0 µg kg$^{-1}$ ADC; (7) Dox + 1.0 µg kg$^{-1}$ ADC; (8) Dox + 0.3 µg kg$^{-1}$ ADC. Doxycycline food was administered to the indicated group from day 10 to day 36. Tumor growth was monitored by the IVIS system after luciferin injection for 15 min. Loss of body weight during the course of the study was also monitored. At the end of the studies mice were killed and tumors extracted, weighed and fixed in formalin solution and processed for sectioning and immunohistochemistry staining.

**In vitro transcription assay**. In vitro transcription reactions were set up as described above and the nuclear extraction was purified from DU145 cells using Nuclear Extraction Kit (Abcam). The protein concentration of each sample was determined by BCA Protein Assay Kit (Thermo). Briefly, equal amount of nuclear extraction was mixed with the in vitro transcription buffer, linear DNA templates either pEGFP-N1 or HeLaScribe Positive control DNA (HS DNA) following the manufacturer's recipe. A standard reaction containing no NTPs was used as a "negative transcription control". The resulting RNA was purified using RNeasy Micro Kit (Qiagen) and BaselineZero DNase Kit (Epicenter), and was subjected to reverse transcription using Sensiscript RT kit (Qiagen). After reaction, 1 µl of the reverse transcription mix was used for either standard PCR amplification or qPCR to visualize the quantity of the in vitro transcribed RNA product. The primers for generating the linear DNA are:

HS-DNA:
5′-CTCATGTTTGACAGCTTATCGATCCGGGC-3′;
5′-ACAGGACGGGTGTGGTCGCCATGAT-3′.
The primer-Probe mix for detection of HSDNA transcript by qPCR are
Primers: 5′-GCCGGGCCTCTTGCGGGGATAT-3′ and 5′-CGGCCAAAGCGGTCGGACAGT-3′; Probe: 5′-FAM-TGGCGTGCTGCTAGCGCTAT-BHQ3′;

**In vivo ubiquitination assay**. The ubiquitination assays were modified from a previous study[63]. DU145 cells were transiently co-transfected with indicated plasmids. Forty-eight hours after transfection cells were treated with 5 µg ml$^{-1}$ MG132 (Sigma) for 12 h. Cells were then harvested and lysed in denaturing buffer (6 M guanidine-HCl, 0.1 M Na$_2$HPO$_4$/NaH$_2$PO$_4$ and 10 mM imidazole). The cell lysates were incubated with nickel beads for 3 h, washed and immunoblotted with indicated antibodies.

**Transcription assay**. Nascent RNA was labeled by 5-ethynyl uridine (5-EU) incorporation and was evaluated by using the Click-iT RNA Imaging Kit

(Invitrogen)[39]. RBX1 or control shRNA expressed cells were indicated by tRFP signal. 5-EU signal, RFP and DAPI staining were captured using the Leica imaging system. Images were processed by CellProfiler Software. DAPI staining was used to define the area of the nuclei, and the fluorescence signal intensity of 5-EU in every tRFP expressed cells was quantified for each nucleus.

**Statistical analysis**. Each experiment was repeated twice or more, unless otherwise noted. No samples or animals were excluded from the analysis. For the mouse experiment, no statistical method was used to predetermine sample size. The samples or animals were randomly assigned to different treatment groups. A laboratory technician who provided animal care and collected data was blinded to the group allocation during all animal experiments and outcome assessment. Differences between two groups were analyzed by the Student's $t$-test. One-way ANOVA followed by Tukey's $t$-test was conducted to compare three or more groups of independent samples. A $p$ value < 0.05 was considered statistically significant.

## Data availability
All relevant data are available from authors upon request.

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

## Acknowledgements

We thank E. Lander and D. Sabatini for providing the Cas9 expressing vector and CRISPR pooled library. We thank L. Huang and M. Bar-Eli for lentivirus production. We also thank J. Wang for technical support in prostate orthotopic tumor experiments. The following cores were used: MD Anderson Flow Cytometry and Cellular Imaging Core, MD Anderson Clinical Pathology, Veterinary Medicine and Surgery Core, MD Anderson Sequencing and Microarray Core, ICBM Core Facility at IUSM. This work was supported in part by US National Institutes of Health grants R01CA203737 (X.L.), R01CA206366 (X.H. and X.L.), R01CA211861 (B.H.), and R21CA213535 (J.W. and X.L.), and by Indiana University Strategic Research Initiative fund (X.L.), Vera Bradley Foundation for Breast Cancer Research (X.L. and X.Z.) and American Cancer Society Institutional Research Grant (Yunhua Liu).

## Author contributions

Y.L. (Yujing Li), Y.L. (Yunhua Liu), Y.L. (Yunlong Liu), K.H., B.H., Y.D. and X.L. designed the experiments, X.Z., G.J. and X.L. supervised the study, Y.L. (Yujing Li), Y.L. (Yunhua Liu), H.X., K.V.J., Y.F., Z.Z., L.Z., and L.W. conducted the experiments, J.W. provided antibody-drug conjugates, G.J., Z.L., Y.L. (Yujing Li), M.F., M.R., B.P.S. and X.H. analyzed the data, X.L., Y.L. (Yujing Li), and Y.L. (Yunhua Liu) wrote the manuscript, and X.L. edited the manuscript.

## Additional information

**Competing interests:** The authors declare no competing interests.

