## [Peer Review File · Nature Communications]

Reviewers' Comments:

Reviewer #1:

Remarks to the Author:

In this manuscript, Li et al. provide convincing evidence that heterozygous deletion of chromosome 17p uniquely sensitizes prostate tumor cells to RBX1 loss and RNA polymerase II inhibition. Li et al. first analyze patient TCGA and SU2C cohorts to show that the frequently-occurring 17p heterozygous chromosome deletion encompasses TP53 as well as POLR2A (catalytic RNA pol II subunit) deletion, and is associated with advanced tumors. The authors then perform a CRISPR screen in isogenic DU145 cells with or without heterozygous 17p loss to identify synthetic lethality genes, and focus on RBX1, an E3 ligase. Further validation in multiple prostate cancer cell lines with endogenous 17p heterozygous deletion (PC-3, VCaP) revealed that cells with heterozygous 17 loss are selectively sensitive to RBX1 loss. This finding was also confirmed in both DU145, 22RV1, and VCaP xenograft models. The authors uncover an underlying mechanism to explain this synthetic lethality and show in DU145 cells that RBX1 can K63 poly-ubiquitinate POLR2A, which leads to reduced mRNA transcription. Based on these findings, the authors demonstrate that combination RBX1 knockdown and POLR2A inhibition with an alpha-amanitin antibody drug conjugate can effectively and selectively block tumor cell growth in cells with 17p heterozygous loss, using both cell line and xenograft studies. Throughout the study, the authors provide appropriate controls and methods to demonstrate their findings, and the vast majority of the data presented in the figures is well-labeled and easy to understand. This study represents a large, comprehensive effort with findings that would be very beneficial for prostate tumor subtype characterization and selection of optimal therapies.

This study is of great interest to the prostate cancer field and reveals a few novel findings: 1) RBX1 can K63 polyubiquitinate RNA pol II, a modification which can increase RNA pol II activity; 2) combination of RBX1 loss and RNA pol II inhibition can selectively block 17 heterozygous loss tumor cell growth. However, the authors point out that POLR2A loss in 17p heterozygous loss tumors and the subsequent susceptibility to RNA pol II inhibitors has also been shown in colorectal cancer. While this strengthens the rationale of the study, it does slightly reduce the overall novelty.

Some key points that remain to be addressed by the authors:

1. The fluency of the English language could be improved particularly in the abstract, introduction, and discussion sections. There are also a few spelling errors in key headings.
2. Exact p-values should be indicated in all figures where statistical analyses were performed. Additionally, p-values are missing from the following: Figure 3c; 4a, c, d; 6a, c, d; 7b, c, e, f; 8b and c (if relevant), d. Supplemental Figure 2a, b; 5b; 7c, d.
3. The first reference could be updated to a paper with more recent prostate cancer statistics than the current paper from 2014. (Example: Siegel et al. 2018.)
4. It is not clear why the authors chose to focus solely on RBX1 for their downstream analysis and not GTF2HI. The authors mention a slightly less significant effect of GTF2HI knockdown, but additional discussion of this point (such as, perhaps the function of GTF2HI is less cancer cell relevant?) could be useful.
5. Could the authors explain the rationale for using EpCAM antibodies for their alpha-amanitin ADC?
6. Specific figure notes:
Figure 1 – Figure 1f as called out in the text is missing from both Figure 1 and the figure legend.

This missing data is important for the authors' rationale to conclude that 17p loss is associated with and important during tumor progression. How did the authors determine the changes in 17p loss between tumor grades were significant?

Figure 3 – In figure 3h, the level of RBX1 rescue is higher than the normal endogenous levels of the protein. Is this additional increase beyond endogenous levels physiologically relevant?

Figure 6 – Could the authors perform the EU incorporation assay in PC-3 or VCaP cells with the endogenous 17p heterozygous loss?

Figure 7 – The authors show IC50 curves in panels b, c, e, f but only mention the exact IC50 value for one cell line in panel b. This information would be useful for comparison purposes and could be provided in a supplementary table.

Figure 8 – The labeling of Figure 8b and c could be more clear.

Reviewer #2:

Remarks to the Author:

1) Summary

The paper by Li et al. focused on understanding the role of chromosome 17p, whose mutation or deletion is frequently found in a majority of human tumors. This region contains ~200 protein-coding genes (including tumor suppressor p53 and RNA polymerase II catalytic subunit [POLR2A]) and non-coding RNA genes, as well as additional regulatory elements. The authors have used a castration-resistant prostate cancer (CRPC) cell DU145 to create a line in which one copy of 17p has been deleted. In a CRISPR-based screen, the authors have discovered that the loss of RBX1 leads to selective killing of 17p-deficient cells.

RBX1 is a RING finger protein that constitutes a key component of the multi-subunit Cullin-RING E3 ubiquitin ligase complexes (CRL). The authors provide biochemical evidence suggesting that RBX1 acts to modify POLR2A by K63-polyubiquitination and is critical for POL II transcription. Moreover, depletion of RBX1 appears to sensitize 17p-deficient cells to POLR2A inhibition. Based on these data, the authors suggest RBX1 as a potential vulnerability augmented by 17p deletion in human CRPC.

2) Critique

a) CRL is the largest E3 ubiquitin ligase family of > 200 members, about half of E3s found in humans. RBX1 and close homologue RBX2 are the common component of all CRLs. RBX1/RBX1 functions to recruit E2 ubiquitin conjugating enzyme for catalyzing the transfer of ubiquitin to a bound substrate, whereas a substrate is bound to E3 CRL via protein-protein interactions with CRL's distinct substrate binding receptors (such as F-box proteins for CRL1).

The biochemical characterization of the RBX1 effects in this work (Figs. 5-8) primarily focused on RBX1 alone by depletion or over-expression approaches. Given numerous cellular proteins (and hence biological processes) affected by such approaches, it is highly doubtful that the current study would yield any specific insights. The authors need to determine specific E3 CRL(s) responsible for the RBX1 effects observed.

b) The reported RBX1 effects on POLR2A by K63-polyubiquitination (Fig. 5) is inconsistent with literature. In the published work by Ribar et al. (2007, MCB 27 [8]: 3211-6), Pol II catalytic subunit was found modified by K48-polyubiquitination, which leads to degradation. These findings

contradicted observations reported in the current work.

c) The major finding that RBX1 is ONLY essential for proliferation in 17p-deicient cells is inconsistent with a large body of work on RBX1 in the literature. RBX1 was shown to be an essential gene required for mouse embryonic development (Tan et al, 2009, PNAS, 106: 6203-8). In addition, CRISPR-based genome-wide analysis shows that RBX1 is an essential gene in all four cell lines tested (KBM7, K562, Jiyoye, and Raji) (Wang et al., 2015, Science, 350: 1096-1101).

In all, the concerns/weaknesses noted above have significantly weakened the major premise of this paper suggesting that RBX1 is a candidate vulnerability gene in 17p-deletion cells.

Reviewer #3:

Remarks to the Author:

Li and colleagues describe in their study a genetic survey for vulnerabilities in prostate cancers with deletions of chromosome 17p, the home of p53.

The principle findings are: (1) Identification of POLR2A as an actionable vulnerability in 17p deleted prostate cancers, based on the group's previous work in colon cancer. (2) Discovery of RBX1 and GTF2H1 as novel targets through a CRISPR dropout screen (which is however does not seem to be genome-wide, as stated in the abstract!). (3) Identification of direct POLR2A activation through RBX1, including a large number of mechanistic insights (4) Demonstration of synergistic effects of POLR2A and RBX1 inhibition for prostate cancer targeting.

The study presents a large body of work. More than 60% of prostate cancers has 17p alterations. Identifying therapeutic vulnerabilities in this context is thus highly relevant. The experiments are well controlled. The paper is well written and discussed. The manuscript would however benefit from some further revision before publication.

1. Prostate cancer genomics

Figure 1A:

- Cohort size should be indicated as n = 492 instead of n = 499 (TCGA mutation and copy number data is only available from 492 prostate tumours) (please also correct in figure legend)
- Within the TCGA dataset, there are some prostate tumours showing a deep deletion (described as "homozygous" in cBioPortal) plus a TP53 point mutation (with VAFs ranging from 0.3 to 0.8). This seems to be counterintuitive. Please comment in the figure legend.

Figure 1D: prostate tumours are categorized as "T2", "T3" and "T4" pathological stages. Please add the Gleason score, if available.

Figure 1F is missing (page 5, first section). It should show TP53 mutations and deep deletions in metastatic prostate cancer according to text.

2. CRISPR/Cas dropout screen

Limitations:

- Only one cell line (DU145) is used for the screen. Inclusion of more cell lines might could potentially identify further vulnerabilities.
- In the abstract it is stated that the screen is genome-wide. In the results and materials and methods however, only a focused screen is described (3700 genes; as stated in the results and materials and methods).
- What was the rationale to use the specific gRNA library against nuclear proteins?

Supplementary Figure 1B: change the order of the indicated band sizes: 474 bp should be listed above 469 bp

Methods:

- "A single colony with decent fold-change in Cas9 expression (...)": Vague description, please specify (page 16, third section).
- Please specify the MOI used for infection of cells.

3. RBX1 in vitro depletion experiments

Please change "green fluorescence protein (RFP)" to "red fluorescence protein (RFP)" (page 6, second section).

The authors might consider commenting on why they choose to use shRNA mediated gene knockdown rather than CRISPR/Cas mediated knockout?

4. RBX1 in vivo depletion experiments

Why did the authors choose not to use parental and 17ploss DU145 cells for orthotopic transplantation experiments?

5. Functional analysis of RBX1

Results/data from mass spectrometry are/is missing

Figure 5C:

- Knockdown of RBX1 using shRBX1 #2 and shRBX1 #3 seems to be rather weak (Western blot). However, the ubiquitination level of POLR2A is markedly reduced. Moreover, Supplementary Figure 2A shows that the knockdown efficiencies of shRBX1 #1 and shRBX1 #2 seem to be a comparable in DU145 cells in the initial experiments. Please comment.

Response to Reviewers

Attached for consideration for publication as a Research Article in *Nature Communications* is our revised manuscript entitled, “**Heterozygous Deletion of Chromosome 17p Renders Prostate Cancer Vulnerable to the Inhibition of RNA Polymerase II**”. We thank you for your helpful comments and critiques. In the revised manuscript, we believe that we have addressed the reviewers’ concerns arising from the initial manuscript. Additional data has been added into the manuscript and some of the main and supplementary figures have been modified to include new results. I have responded to the review comments on a point-by-point basis as below:

Reviewer #1

1. *“Throughout the study, the authors provide appropriate controls and methods to demonstrate their findings, and the vast majority of the data presented in the figures is well-labeled and easy to understand. This study represents a large, comprehensive effort with findings that would be very beneficial for prostate tumor subtype characterization and selection of optimal therapies.”*

We thank Reviewer#1 for his/her positive comments and helpful suggestions. In the revised manuscript, we include a significant number of additional data (Fig. 1f; Fig. 3g-h; Fig. S4b-e; Fig. S5a; Fig. S7; Fig. S9; Table S2 and S3) to further support our conclusion.

2. *“The fluency of the English language could be improved particularly in the abstract, introduction, and discussion sections. There are also a few spelling errors in key headings.”*

We agree with the reviewer and have rewritten the abstract, introduction and discussion sections, and all typos have been corrected as well.

3. *“Exact p-values should be indicated in all figures where statistical analyses were performed. Additionally, p-values are missing from the following...”*

We thank the review for pointing out this issue. All the statistical evaluation has been added as suggested for multiple comparisons in the figures. Unless otherwise noted, data are presented as mean \pm SD, and Student’s t-test (unpaired, two-tailed) was used to compare two groups of independent samples. In the unpaired t-test, we assumed equal variance and that no samples were excluded from the analysis. One-way ANOVA followed by Tukey’s t test was conducted to compare three or more groups of independent samples. $p < 0.05$ was considered statistically significant.

4. *“The first reference could be updated to a paper with more recent prostate cancer statistics than the current paper from 2014.”*

We agree with the reviewer and have updated the reference as suggested.

5. *“It is not clear why the authors chose to focus solely on RBX1 for their downstream analysis and not GTF2HI. The authors mention a slightly less significant effect of GTF2HI knockdown, but additional discussion of this point (such as, perhaps the function of GTF2HI is less cancer cell relevant?) could be useful.”*

We thank the review for this suggestion. GTF2H1, a subunit of the transcription factor IIIH (TFIIH), was reported to play an important role in tumorigenesis by modulating nucleotide excision repair process and transcriptional activity via its interaction with a variety of factors. However, targeting the transcription factors has been hampered by its poor druggability. In contrast, RBX1 possesses several ideal features and has been identified as anticancer target recently (Wei, D et al, 2010, Genes Cancer, 1:700-7; Jia, L et al, 2009, Cancer Res; 69:4974-82). First, it's a “druggable” target. Second, RBX1 is overexpressed in human cancers including lung, liver, breast colon and ovary, and is also required for cancer cell proliferation as well as for the maintenance of cancer cell phenotypes. Third, normal cells are less sensitive to its inhibition, allowing for potential therapeutic windows. Therefore, we focused on RBX1 for downstream experiments. We have included the discussion into the revised manuscript.

6. *“Could the authors explain the rationale for using EpCAM antibodies for their alpha-amanitin ADC?”*

Clinical application of α -amanitin is hampered by its dose-limiting toxicity in liver through its interaction with the transporting protein OATP1B3, which is exclusively expressed on the membrane of hepatocytes. However, when coupled to proteins, α -amanitin is no longer a substrate for OATP1B3. Therefore, the amanitin-conjugated antibodies have dramatically reduced liver toxicity. In a previous preclinical study, Dr. Moldenhauer and colleagues provided very promising results that anti-EpCAM antibody conjugates with α -amanitin are effective in inhibiting xenografted pancreatic, colorectal, and other types of cancer (Moldenhauer G et al, J Natl Cancer Inst 2012; 104:622-34). In this project, we used anti-EpCAM antibodies to deliver ADCs for CRPC treatment as a proof-of-concept. Most importantly, both 17p-neutral and -loss DU145 cells have robust expression of EpCAM and also show similar rate of antibody internalization as determined by ImageStream flow cytometry (Supplementary Figure 7). Other α -

amanitin-based ADCs have also been developed in the laboratory in collaboration with biotech companies, including α -amanitin-conjugated anti-PSMA antibody.

7. *“Figure 1f as called out in the text is missing from both Figure 1 and the figure legend. This missing data is important for the authors’ rationale to conclude that 17p loss is associated with and important during tumor progression. How did the authors determine the changes in 17p loss between tumor grades were significant?”*

We apologize for our sloppiness and Figure 1f has been now added to both the figure legend and main text. Fisher's exact test was used to determine whether the changes of the 17p loss between tumor grades were significant or not. $p < 0.05$ was considered statistically significant.

8. *“In figure 3h, the level of RBX1 rescue is higher than the normal endogenous levels of the protein. Is this additional increase beyond endogenous levels physiologically relevant?”*

We completely agree with the reviewer that ectopic expression of RBX1 may lead to artifacts when over-expressed at higher levels than endogenous one. Therefore, we redid this experiment with comparable RBX1 expression to its endogenous levels and similar results were observed.

9. *“Figure 6 – Could the authors perform the EU incorporation assay in PC-3 or VCaP cells with the endogenous 17p heterozygous loss?”*

We thank the reviewer for the suggestion. As shown in Supplementary Figure 5a, we performed the nascent mRNA synthesis by EU incorporation assay and found that further depletion of RBX1 in the 17^{loss} cells (PC3) resulted in much more inhibitory effects on the mRNA transcription, as compared to 17^{neutral} cells (22Rv1).

10. *“Figure 7 – The authors show IC50 curves in panels b, c, e, f but only mention the exact IC50 value for one cell line in panel b. This information would be useful for comparison purposes and could be provided in a supplementary table.’*

As suggested, the exact IC50 values of α -amanitin and actinomycin D have been provided in the Supplementary Table 3.

11. *“Figure 8 – The labeling of Figure 8b and c could be more clear.”*

We thank the reviewer for pointing out this concern and the labeling of Figure 8b and c have been updated.

Reviewer #2

1. *“The authors provide biochemical evidence suggesting that RBX1 acts to modify POLR2A by K63-polyubiquitination and is critical for POL II transcription. Moreover, depletion of RBX1 appears to sensitize 17p-deficient cells to POLR2A inhibition. Based on these data, the authors suggest RBX1 as a potential vulnerability augmented by 17p deletion in human CRPC.”*

We thank Reviewer#2 for his/her helpful critiques and suggestions. In the revised manuscript, we include a significant number of additional data (Fig. 1f; Fig. 3g-h; Fig. S4b-e; Fig. S5a; Fig. S7; Fig. S9; Table S2 and S3) to address the reviewers' concerns and further support our conclusion as described below.

2. *“The biochemical characterization of the RBX1 effects in this work (Figs. 5-8) primarily focused on RBX1 alone by depletion or over-expression approaches. Given numerous cellular proteins (and hence biological processes) affected by such approaches, it is highly doubtful that the current study would yield any specific insights. The authors need to determine specific E3 CRL(s) responsible for the RBX1 effects observed.”*

We thank the reviewer for pointing out this concern. Using co-immunoprecipitation and mass spectrometric analyses, we found that POLR2A interacts primarily with the CRL1 and CRL2 E3 ligase complexes through substrate recognition subunit SKP2 or VHL (Fig. S4b). However, we would not exclude the possibility that POLR2A may also interact with other CRL E3 ligase complexes with relatively weaker binding affinity. Depletion of the substrate recognition subunit SKP2, but not VHL, could partially phenocopy the effect of RBX1 knockdown on the K63-associated POLR2A ubiquitination. Additionally, knockdown of SKP2 has notable cell killing effect on the 17p^{loss} DU145 cells, although the effect is not as profound as compared to that of RBX1 (Fig. S4d, e). These results suggest that multiple CRLs E3 ligase complexes, including CRL1, may be involved in the RBX1-mediated effect on the 17p^{loss} cancer cells through their functional interaction with POLR2A or other genes in the 17p deletion region (Fig. S4d, e). We have also discussed the results in the revised manuscript.

3. *“The reported RBX1 effects on POLR2A by K63-polyubiquitination (Fig. 5) is inconsistent with literature. In the published work by Ribar et al. (2007, MCB 27 [8]: 3211-6), Pol II catalytic subunit was*

found modified by K48-polyubiquitination, which leads to degradation. These findings contradicted observations reported in the current work.”

We agree with the reviewer that the catalytic subunit of the RNA Pol II complex, POLR2A, was subjected to DNA damage-induced degradation through K48-polyubiquitination. Additionally, it was also reported that ubiquitination of POLR2A also occurs on the lysine 63 of ubiquitin, suggesting a non-degradative signaling role of K63 ubiquitination during transcriptional regulation (Sharp PA et al, 2004, *Biochemistry*, 43: 15223-99). Various ubiquitination events appear to function in the contexts of internal and external changes in the cell.

4. “The major finding that RBX1 is ONLY essential for proliferation in 17p-decient cells is inconsistent with a large body of work on RBX1 in the literature. RBX1 was shown to be an essential gene required for mouse embryonic development (Tan et al, 2009, PNAS, 106: 6203-8). In addition, CRISPR-based genome-wide analysis shows that RBX1 is an essential gene in all four cell lines tested (KBM7, K562, Jiyoye, and Raji) (Wang et al., 2015, Science, 350: 1096-1101).”

We thank the reviewer for pointing out this concern. We also noticed these previous reports in the literature. While the genes essential for cell survival are nonexclusively essential for embryonic development, many developmentally essential genes are not essential for single cell survival. RBX1 is such a case. In 2009, RBX1 was reported to possess several ideal features as anticancer target: 1) It is a “druggable” enzyme and is overexpressed in many types of human cancer including lung, liver, breast colon and ovarian cancer. RBX1 promotes cancer cell proliferation as well as contributes to the maintenance of cancer cell phenotype. Therefore, normal cells are less sensitive to the inhibition of RBX1, creating potential therapeutic windows (Wei, D et al, 2010, *Genes Cancer*, 1:700-7; Jia, L et al, 2009, *Cancer Res*; 69:4974-82); 2) Although RBX1 and its paralog RBX2 (also named as RNF7) only share 53% of overall sequence identity, 7 of 8 Cys/His residues that constitute their RING finger domain are identical. Both RBX1 and RBX2 are found to be evolutionarily conserved among many species from yeast to humans. More importantly, either human RBX1 or RBX2 can rescue the lethal phenotype caused by deletion of the RBX ortholog Hrt1 in yeast that contains only one family member, indicating they are functionally equivalent at least in yeast (Swaroop M et al, 2000, *Oncogene*, 19:2855-66); 3) The tissue expression patterns of human or mouse RBX1 and RBX2 are similar. In humans, both proteins are ubiquitously expressed with much higher expression in the heart, skeletal muscle, and testes, and less expression in the brain, lung, kidney, and placenta, indicating they may functionally replaceable to some extent (Duan H et al, 1999, *Mol Cell Biol*, 9:3145-55).

We also analyzed the genomes of the four cell lines in the cited paper using Cancer Cell Line Encyclopedia (CCLE) data base. Interestingly, KBM7 is a haploid cell line that only contains one copy of Chr17p, and K562 is a cell line with heterozygous loss of 17p. Both of them are assumed to be sensitive to the inhibition of RBX1. For the other two cell lines, the genomic data of Jiyoye cell line is not available from CCLE and the literature. Raji cell line seems to contain neural copy number of 17p, whose sensitivity is probably due to the unidentified genetic reason. Most importantly, our analysis of CCLE data base revealed that 11 cell lines contain **homozygous** deletion of RBX1, including ACHN, HT115, KALS-1, LCLC-103H, MSTO-211H, REC-1, SH-SY5Y, SK-CO-1, SK-N-SH, SNU-719, and TE-1 cell lines, indicating that RBX1 is not an essential gene for global cell survival.

Reviewer #3

1. "The study presents a large body of work. More than 60% of prostate cancers has 17p alterations. Identifying therapeutic vulnerabilities in this context is thus highly relevant. The experiments are well controlled. The paper is well written and discussed. The manuscript would however benefit from some further revision before publication."

We thank the reviewer for his/her positive comment and helpful suggestions. In the revised manuscript, we now include a significant number of additional data (Fig. 1f; Fig. 3g-h; Fig. S4b-e; Fig. S5a; Fig. S7; Fig. S9; Table S2 and S3) to further support our conclusion.

2. "Cohort size should be indicated as $n = 492$ instead of $n = 499$. Within the TCGA dataset, there are some prostate tumours showing a deep deletion plus a TP53 point. This seems to be counterintuitive. Please comment in the figure legend."

We thank the reviewer for pointing out this error and have corrected it accordingly. Gene copy number levels are derived from the copy-number analysis algorithms GISTIC or RAE, and indicate the copy-number level per gene in the analyzed tumor tissue. Due to the intra-tumor heterogeneity, it is likely that one tumor tissue contains tumor cells from two different colonies with homozygous deletion of TP53 or mutant TP53. We have commented in the figure legend.

3. "Figure 1D: prostate tumours are categorized as "T2", "T3" and "T4" pathological stages. Please add the Gleason score, if available."

The Gleason score has been provided as suggested and Fisher's exact test was used to determine whether the changes in 17p loss between tumor grades were significant or not. $p < 0.05$ was considered statistically significant.

4. *“Figure 1F is missing (page 5, first section). It should show TP53 mutations and deep deletions in metastatic prostate cancer according to text.”*

We apologize for our sloppiness. Figure 1f has now been added to Figure 1.

5. *“Only one cell line (DU145) is used for the screen. Inclusion of more cell lines might could potentially identify further vulnerabilities. In the abstract it is stated that the screen is genome-wide. In the results and materials and methods however, only a focused screen is described (3700 genes; as stated in the results and materials and methods). What was the rationale to use the specific gRNA library against nuclear proteins?”*

We completely agree with the reviewer that screens from multiple cell lines could potentially identify further vulnerabilities. Knockout of a fragment of ~20 megabases from Chr17p has been technically challenging and very time-consuming. While we only used the isogenic pair of DU145 cell lines for screen, the screen results have been tested in a number of cell lines harboring neutral and hemi-loss 17p. In the meanwhile, we are now still optimizing CRISPR-Cas9 knockout assay to more efficiently knock out large DNA fragments. Ideally, the whole genome gRNA library would provide more potential targets in the screen. However, based on our experience, a focused screen seems to provide more reliable targets with fewer false-positives. A number of important genes (such as TP53, POLR2A, EIF5A) in the 17p encode nuclear proteins, which made us first identify targets in the nucleus that potentially interact with those key players. We will certainly plan to conduct a genome-wide screen in the near future.

6. *“Supplementary Figure 1B: change the order of the indicated band sizes. ... A single colony with decent fold-change in Cas9 expression... Vague description, please specify. Please specify the MOI used for infection of cells.”*

We agree with the reviewer that the term “decent expression” may not be accurate and now we have changed it to “robust expression (>50-fold induction)”. We also switched the DNA marker labelling and included the MOI (~0.3) in the methods.

7. *“Please change “green fluorescence protein (RFP)” to “red fluorescence protein (RFP)” (page 6, second section). The authors might consider commenting on why they choose to use shRNA mediated gene knockdown rather than CRISPR/Cas9 mediated knockout?”*

We apologize for our sloppiness and the error has been corrected. In 2009, RBX1 was reported to possess several ideal features as anticancer target. As a “druggable” target, RBX1 is overexpressed in many human cancers including lung, liver, breast colon and ovarian cancer. It is required for cancer cell proliferation as well as for the maintenance of cancer cell phenotype. Therefore, normal cells are less sensitive to the RBX1 silencing, creating potential therapeutic windows (Wei, D et al, 2010, Genes Cancer, 1:700-7; Jia, L et al, 2009, Cancer Res; 69:4974-82). In this project, we used dox-induced knockdown of RBX1 for CRPC treatment as a proof-of-concept. In the recent years, we have been developing nanoparticles for highly efficient delivery of small RNAs in vivo (Wang H et al, Adv Mater, 28(2):347-355; Liu Y et al, Nature, 520(7549):697-701). We believe that improved RNA nanoparticles would have great clinical potential for cancer therapy.

8. *“Why did the authors choose not to use parental and 17ploss DU145 cells for orthotopic transplantation experiments?”*

Cell sensitivity to the inhibition of RBX1 may be due to specific genetic background of DU145 cells. Since we have tested the isogenic pair of DU145 cells in the xenograft models (Fig. 4a, b, subcutaneous injection), we wanted to confirm whether 17p loss confers sensitivity to the RBX1 inhibition in cell lines with different genetic background in orthotopic tumor models.

9. *“Results/data from mass spectrometry are/is missing”*

As suggested, the mass spectrometry data have been included in Supplementary Table 3.

10. *“Figure 5C: Knockdown of RBX1 using shRBX1 #2 and shRBX1 #3 seems to be rather weak (Western blot). However, the ubiquitination level of POLR2A is markedly reduced. Moreover, Supplementary Figure 2A shows that the knockdown efficiencies of shRBX1 #1 and shRBX1 #2 seem to be a comparable in DU145 cells in the initial experiments. Please comment.”*

In Figure 5C, we mistakenly used wrong Western blotting results. Both shRBX1 and shRBX2 efficiently

knocked down the expression of RBX1. To ensure the data accuracy, we redid all the experiment included in Figure 5C. The figure now includes the new results.

Reviewers' Comments:

Reviewer #1:

Remarks to the Author:

The authors have adequately addressed all my concerns.

Reviewer #2:

Remarks to the Author:

The revised paper by Li et al. was improved. However, substantial concerns remain as detailed below.

Original comment point 1: CRL is the largest E3 ubiquitin ligase family of > 200 members, about half of E3s found in humans. RBX1 and close homologue RBX2 are the common component of all CRLs. RBX1/RBX2 functions to recruit E2 ubiquitin conjugating enzyme for catalyzing the transfer of ubiquitin to a bound substrate, whereas a substrate is bound to E3 CRL via protein-protein interactions with CRL's distinct substrate binding receptors (such as F-box proteins for CRL1).

The biochemical characterization of the RBX1 effects in this work (Figs. 5-8) primarily focused on RBX1 alone by depletion or over-expression approaches. Given numerous cellular proteins (and hence biological processes) affected by such approaches, it is highly doubtful that the current study would yield any specific insights. The authors need to determine specific E3 CRL(s) responsible for the RBX1 effects observed.

Authors' response: We thank the reviewer for pointing out this concern. Using co-immunoprecipitation and mass spectrometric analyses, we found that POLR2A interacts primarily with the CRL1 and CRL2 E3 ligase complexes through substrate recognition subunit SKP2 or VHL (Fig. S4b). However, we would not exclude the possibility that POLR2A may also interact with other CRL E3 ligase complexes with relatively weaker binding affinity. Depletion of the substrate recognition subunit SKP2, but not VHL, could partially phenocopy the effect of RBX1 knockdown on the K63-associated POLR2A ubiquitination. Additionally, knockdown of SKP2 has notable cell killing effect on the 17ploss DU145 cells, although the effect is not as profound as compared to that of RBX1 (Fig. S4d, e). These results suggest that multiple CRLs E3 ligase complexes, including CRL1, may be involved in the RBX1-mediated effect on the 17ploss cancer cells through their functional interaction with POLR2A or other genes in the 17p deletion region (Fig. S4d, e). We have also discussed the results in the revised manuscript.

Reviewer's new comments: While I do appreciate authors' efforts in revision, the new data, however, is far from satisfactory. First, the effects of Skp2 knockdown in selective cell killing are marginal (revised Fig. S4e). They are not very different from the shNT control shown in Fig. 3a. Second, the biochemical evidence for a direct role by Skp2 in targeting POLR2A for ubiquitination is insufficient. Typically, experiments are needed to show that Skp2 binds to POLR2A directly. Reconstitution experiments will be needed to show that SCF (Skp2) mediates the ubiquitination of POLR2A.

Unfortunately, the relatively weak effects of E3 SCF (Skp2) on POLR2A raises substantial concerns regarding the authors' main hypothesis suggesting a prominent role of ubiquitination of POLR2A in the pathophysiological effects of chromosome 17p-deletion.

Original comment point 2: The major finding that RBX1 is ONLY essential for proliferation in 17p-decient cells is inconsistent with a large body of work on RBX1 in the literature. RBX1 was shown to be an essential gene required for mouse embryonic development (Tan et al, 2009, PNAS, 106: 6203-8). In addition, CRISPR-based genome-wide analysis shows that RBX1 is an essential gene in

all four cell lines tested (KBM7, K562, Jiyoye, and Raji) (Wang et al., 2015, Science, 350: 1096-1101).

Authors' response: We thank the reviewer for pointing out this concern. We also noticed these previous reports in the literature. While the genes essential for cell survival are nonexclusively essential for embryonic development, many developmentally essential genes are not essential for single cell survival. RBX1 is such a case. In 2009, RBX1 was reported to possess several ideal features as anticancer target: 1) It is a "druggable" enzyme and is overexpressed in many types of human cancer including lung, liver, breast colon and ovarian cancer. RBX1 promotes cancer cell proliferation as well as contributes to the maintenance of cancer cell phenotype. Therefore, normal cells are less sensitive to the inhibition of RBX1, creating potential therapeutic windows (Wei, D et al, 2010, Genes Cancer, 1:700-7; Jia, L et al, 2009, Cancer Res; 69:4974-82); 2) Although RBX1 and its paralog RBX2 (also named as RNF7) only share 53% of overall sequence identity, 7 of 8 Cys/His residues that constitute their RING finger domain are identical. Both RBX1 and RBX2 are found to be evolutionarily conserved among many species from yeast to humans. More importantly, either human RBX1 or RBX2 can rescue the lethal phenotype caused by deletion of the RBX ortholog Hrt1 in yeast that contains only one family member, indicating they are functionally equivalent at least in yeast (Swaroop M et al, 2000, Oncogene, 19:2855-66); 3) The tissue expression patterns of human or mouse RBX1 and RBX2 are similar. In humans, both proteins are ubiquitously expressed with much higher expression in the heart, skeletal muscle, and testes, and less expression in the brain, lung, kidney, and placenta, indicating they may functionally replaceable to some extent (Duan H et al, 1999, Mol Cell Biol, 9:3145-55).

We also analyzed the genomes of the four cell lines in the cited paper using Cancer Cell Line Encyclopedia (CCLE) data base. Interestingly, KBM7 is a haploid cell line that only contains one copy of Chr17p, and K562 is a cell line with heterozygous loss of 17p. Both of them are assumed to be sensitive to the inhibition of RBX1. For the other two cell lines, the genomic data of Jiyoye cell line is not available from CCLE and the literature. Raji cell line seems to contain neural copy number of 17p, whose sensitivity is probably due to the unidentified genetic reason. Most importantly, our analysis of CCLE data base revealed that 11 cell lines contain homozygous deletion of RBX1, including ACHN, HT115, KALS-1, LCLC-103H, MSTO-211H, REC-1, SH-SY5Y, SK-CO-1, SK-N-SH, SNU-719, and TE-1 cell lines, indicating that RBX1 is not an essential gene for global cell survival.

Reviewer's new comments: The authors appear to challenge a widely accepted view in the E3 CRL field that RBX1 is an essential gene. The did raise an intriguing idea that RBX1 lethality may be related to heterozygous loss of 17p such as K562 cell line. I would suggest that the authors prepare a supplement figure to make this point in the manuscript.

I do have a suggestion for the authors to determine whether KBM7 and K562 cells resemble 17p-deficient DU145 cells such that depletion of RBX1 sensitizes KBM7 and K562 cells to POLR2A inhibition specifically.

Overall, while I do find authors' findings intriguing, however, it is my opinion that the authors do not have a cohesive set of convincing data to present a sound, novel mechanistic idea that helps explain the link between chromosome 17p deletion and cancer.

Reviewer #3:

Remarks to the Author:

The changes and additions made by the authors have substantially improved the quality of the revised manuscript, with respect to the screening and genetics aspects. One important correction, however, still needs to be made. The authors used a CRISPR library with sgRNAs targeting genes

encoding nuclear proteins. This library is NOT genome-wide. The authors should replace "genome-wide CRISPR-Cas9 screen" in the abstract with "focused CRISPR-Cas9 screen").

Response to Reviewers

Attached for consideration for publication as a Research Article in *Nature Communications* is our revised manuscript entitled, “**Heterozygous Deletion of Chromosome 17p Renders Prostate Cancer Vulnerable to the Inhibition of RNA Polymerase II**”. We thank you for your helpful comments and critiques. We are glad that we have addressed most of the reviewers’ concern. In the revised manuscript, we have done additional experiments to address the remaining concerns. Additional data has been added into the manuscript and some of the supplementary figures have been modified to include new results. I have responded to the review comments on a point-by-point basis as below:

Reviewer #1

1. *“The authors have adequately addressed all my concerns.”*

We thank Reviewer#1 for his/her positive comments and helpful suggestions during the review process.

Reviewer #2

1. *“Reviewer’s new comments: While I do appreciate authors’ efforts in revision, the new data, however, is far from satisfactory. First, the effects of Skp2 knockdown in selective cell killing are marginal (revised Fig. S4e). Second, the biochemical evidence for a direct role by Skp2 in targeting POLR2A for ubiquitination is insufficient. Typically, experiments are needed to show that Skp2 binds to POLR2A directly.” Unfortunately, the relatively weak effects of E3 SCF (Skp2) on POLR2A raises substantial concerns regarding the authors’ main hypothesis suggesting a prominent role of ubiquitination of POLR2A in the pathophysiological effects of chromosome 17p-deletion.”*

We thank the reviewer for pointing out this concern. We agree with the reviewer that the SKP2 knockdown alone has relatively weaker effects on the POLR2A ubiquitination and 17p^{loss} cell proliferation when compared to those of RBX1 depletion, which we have acknowledged in the revised manuscript. Using co-immunoprecipitation and mass spectrometric analyses, we found that POLR2A/RBX1 not only interacts with CRL1 and 2 E3 ligase complexes, but also interacts with CRL3 and 7, although at a relatively weaker binding affinity (Supplementary Fig. 4b). Nevertheless, depletion of the substrate recognition subunit SKP2 could partially phenocopy the effect of RBX1 knockdown on the K63-associated POLR2A ubiquitination (Supplementary Fig. S4d). The weaker effect of SKP2, compared to that of RBX1,

suggest that POLR2A may indeed interact with other CRL E3 ligases in addition to SKP2. We have revised the text to include it.

To validate the direct interaction of POLR2A with SKP2, we have conducted the following additional assays: 1) Using reverse co-immunoprecipitation, we verified that POLR2A/RBX1 and SKP2 are in the same protein complex (Supplementary Fig. 4b, right panel); 2) We further confirmed the direct interaction of POLR2A with SKP2 using pull-down assays. We observed that purified GST-SKP2 was able to pull down a considerable amount of POLR2A compared to the well-documented POLR2A interaction protein POLR2H, indicating the direct interaction of SKP2 with POLR2A (Supplementary Fig. 4e). 3) Using confocal single plane microscopy, we found that POLR2A co-localized significantly with RBX1 or SKP2 in the nucleus, with Pearson correlation coefficient of 0.76 and 0.44, respectively (Supplementary Fig. 4f). 4) We also found that overexpression of SKP2 significantly increased the levels of ubiquitinated POLR2A in the presence of HA-tagged ubiquitin. Taken together, these results suggest that SKP2 recognition subunit is involved in the RBX1-mediated effect on the 17p^{loss} cancer cells through their functional interactions with POLR2A in the 17p deletion region.

2. “Reviewer’s new comments: The authors appear to challenge a widely accepted view in the E3 CRL field that RBX1 is an essential gene. They did raise an intriguing idea that RBX1 lethality may be related to heterozygous loss of 17p such as K562 cell line. I would suggest that the authors prepare a supplement figure to make this point in the manuscript. I do have a suggestion for the authors to determine whether KBM7 and K562 cells resemble 17p-deficient DU145 cells such that depletion of RBX1 sensitizes KBM7 and K562 cells to POLR2A inhibition specifically.”

We thank the review for this suggestion. As we mentioned previously, our analysis of CCLE data base revealed that 11 cell lines contain homozygous deletion of RBX1, including ACHN, HT115, KALS-1, LCLC-103H, MSTO-211H, REC-1, SH-SY5Y, SK-CO-1, SK-N-SH, SNU-719, and TE-1 cell lines, indicating that RBX1 is not an essential gene for global cell survival. To address the reviewer’s concern, we first double confirmed the hemizygous loss of 17p in both K562 and KBM7 cell lines using qPCR and Western blot (see attached Fig. S10a, b). As

expected, both cell lines were more sensitive (IC50 ~0.2-0.3 µg/ml) to the treatment of α-amanitin when compared to the 17p^{neutral} chronic myelogenous leukemia (CML) cell lines MEG01 and KU812 (IC50 ~ 1.0-1.5 µg/ml) (see attached Fig. 10c). In addition, knockdown of RBX1 in K562 and KBM7 cells further sensitized them to the α-amanitin-mediated POLR2A inhibition with a 5-fold increase of sensitivity (IC50~ 0.05 µg/ml) (see attached Fig. 10d, e).

3. *“Overall, while I do find authors’ findings intriguing, however, it is my opinion that the authors do not have a cohesive set of convincing data to present a sound, novel mechanistic idea that helps explain the link between chromosome 17p deletion and cancer.”*

In this study, we found that heterozygous deletion of chromosome 17p is a common genomic event in metastatic prostate cancer. We demonstrated that *POLR2A* is included in the 17p deletion region along with *TP53* in a majority of prostate cancers. Inhibition of *POLR2A* with α-amanitin-based ADC selectively suppresses the proliferation, survival and tumor growth of CRPC cells harboring this genomic event. Heterozygous deletion of 17p confers a selective dependence on RBX1, inhibition of which had a synergistic and robust suppression in the growth of CRPC along with the treatment of α-amanitin-conjugated anti-EpCAM antibodies. Given the limited therapeutic options for CRPC, our findings identified potential drug targets from a common Chr17p deletion event in human CRPC. We propose that heterozygous deletion of 17p in human prostate cancer confers therapeutic vulnerabilities, which can be utilized to develop novel targeted cancer therapy for CRPC. We also agree with the reviewer that 17p loss leads to massive change in tumor cells and their interaction with tumor microenvironment. To better understand the biological consequences of 17p loss, we have been studying tumor metastasis, drug resistance, and tumor-immune interaction in the tumors with 17p loss. We believe that we will have better answers for those questions in the near future.

Reviewer #3

1. *“The changes and additions made by the authors have substantially improved the quality of the revised manuscript, with respect to the screening and genetics aspects. One important correction, however, still needs to be made. The authors used a CRISPR library with sgRNAs targeting genes encoding nuclear proteins. This library is NOT genome-wide. The authors should replace “genome-wide CRISPR-Cas9*

screen” in the abstract with “focused CRISPR-Cas9 screen.”

We thank the reviewer for pointing it out. As suggested, we have made this correction in the abstract.

Figure S10. RBX1 depletion sensitized 17p^{loss} chronic myelogenous leukemia cells to POLR2A inhibition. (a) Protein levels of POLR2A and β -Actin in human chronic myelogenous leukemia cells lines. (b) Determination of copy numbers of genes in the proposed deletion region by real-time PCR using gene-specific primers. (c) Cell proliferation of 17p^{neutral} (MEG01 and KU812) and 17p^{loss} cells (K562 and KBM7) treated with different doses of free α -amanitin. (d) Protein levels of RBX1 and β -Actin in K562 and KBM7 cells lines treated with control shRNA (shNT) or RBX1 shRNAs (#1 and #2). (e) K562 and KBM7 cells, with or without RBX1 knockdown, were treated with increasing doses of α -amanitin.

Reviewers' Comments:

Reviewer #2:

Remarks to the Author:

Regrettably, the authors' response to my comments in the previous review cycle is unsatisfactory for reasons detailed below.

Point 1: On the issue of E3 SCF (Skp2)

Reviewer's comments on revision 1: While I do appreciate authors' efforts in revision, the new data, however, is far from satisfactory. First, the effects of Skp2 knockdown in selective cell killing are marginal (revised Fig. S4e). They are not very different from the shNT control shown in Fig. 3a. Second, the biochemical evidence for a direct role by Skp2 in targeting POLR2A for ubiquitination is insufficient. Typically, experiments are needed to show that Skp2 binds to POLR2A directly. Reconstitution experiments will be needed to show that SCF (Skp2) mediates the ubiquitination of POLR2A.

Unfortunately, the relatively weak effects of E3 SCF (Skp2) on POLR2A raises substantial concerns regarding the authors' main hypothesis suggesting a prominent role of ubiquitination of POLR2A in the pathophysiological effects of chromosome 17p-deletion.

Authors' response: We thank the reviewer for pointing out this concern. We agree with the reviewer that the SKP2 knockdown alone has relatively weaker effects on the POLR2A ubiquitination and 17p loss cell proliferation when compared to those of RBX1 depletion, which we have acknowledged in the revised manuscript. Using co-immunoprecipitation and mass spectrometric analyses, we found that POLR2A/RBX1 not only interacts with CRL1 and 2 E3 ligase complexes, but also interacts with CRL3 and 7, although at a relatively weaker binding affinity (Supplementary Fig. 4b). Nevertheless, depletion of the substrate recognition subunit SKP2 could partially phenocopy the effect of RBX1 knockdown on the K63-associated POLR2A ubiquitination (Supplementary Fig. S4d). The weaker effect of SKP2, compared to that of RBX1, suggest that POLR2A may indeed interact with other CRL E3 ligases in addition to SKP2. We have revised the text to include it.

To validate the direct interaction of POLR2A with SKP2, we have conducted the following additional assays: 1) Using reverse co-immunoprecipitation, we verified that POLR2A/RBX1 and SKP2 are in the same protein complex (Supplementary Fig. 4b, right panel); 2) We further confirmed the direct interaction of POLR2A with SKP2 using pull-down assays. We observed that purified GST-SKP2 was able to pull down a considerable amount of POLR2A compared to the well-documented POLR2A interaction protein POLR2H, indicating the direct interaction of SKP2 with POLR2A (Supplementary Fig. 4e). 3) Using confocal single plane microscopy, we found that POLR2A co-localized significantly with RBX1 or SKP2 in the nucleus, with Pearson correlation coefficient of 0.76 and 0.44, respectively (Supplementary Fig. 4f). 4) We also found that overexpression of SKP2 significantly increased the levels of ubiquitinated POLR2A in the presence of HA-tagged ubiquitin. Taken together, these results suggest that SKP2 recognition subunit is involved in the RBX1-mediated effect on the 17p loss cancer cells through their functional interactions with POLR2A in the 17p deletion region.

Reviewer's new comments on revision 2: My first concern "the effects of Skp2 knockdown in selective cell killing are marginal (now revised Fig. S4g). They are not very different from the shNT control shown in Fig. 3a" was NOT addressed by any new data. I remain unconvinced by authors' main hypothesis that E3 SCF (Skp2) has a significant role in the pathophysiological effects of chromosome 17p-deletion.

With respect to my second concern on biochemical evidence, the response is unacceptable. The

revised Fig. S4e is a pull down experiment with crude lysates, not using ALL pure components to properly address direct protein-protein interactions. Also, the percentage of "input" was not indicated, precluding assessment of pull down efficiency. More disturbingly, however, the authors have completely ignored my suggestion to use reconstitution experiments that would demonstrate the ability of E3 SCF (Skp2) to support the ubiquitination of POLR2A.

Point 2: On the issue of RBX1 essentiality for cell growth

I do appreciate authors' new findings in this work that may alert interested investigators to be aware of the complexity concerning RBX1 essentiality. However, the authors have not clearly addressed this issue in the manuscript. I have prepared a paragraph that could be used by the authors in the Discussion:

"Results of CRISPR-based knockout of RBX1 (Fig. 2d) or silencing of RBX1 by shRNA (Fig. 3a, b and Supplementary Fig. 2a) in a castration-resistant prostate cancer (CRPC) cell line DU145 presented in this work have revealed that RBX1 is essential for proliferation only in 17p-deficient cells, but not in 17p-neutral parental cells. These findings suggest that RBX1 is not an essential gene for proliferation at least in DU145 cells. However, RBX1 was shown to be an essential gene required for mouse embryonic development (Tan et al, 2009, PNAS, 106: 6203-8). In addition, CRISPR-based genome-wide analysis shows that RBX1 is an essential gene in all four cell lines tested (KBM7, K562, Jiyoye, and Raji) (Wang et al., 2015, Science, 350: 1096-1101; Blomen et al., 2015, Science, 350: 1092-1095). To understand this discrepancy, we have analyzed RBX1 genetic integrity in a large set of tumor cells using Cancer Cell Line Encyclopedia (CCLE) database. The results revealed homozygous deletion of RBX1 in at least 11 cell lines, including ACHN, HT115, KALS-1, LCLC-103H, MSTO-211H, REC-1, SH-SY5Y, SK-CO-1, SK-N-SH, SNU-719, and TE-1. In addition, KBM7 (used in the Wang study) is a haploid cell line that only contains one copy of Chr17p, while K562 (used in the Wang study) is a cell line with heterozygous loss of 17p. Of note, the genomic data of Jiyoye cell line (used in the Wang study) is unavailable from CCLE. Raji cell line (used in the Wang study) seems to contain neural copy number of 17p. Collectively these data suggest that the essentiality of RBX1 for cell growth may be context-dependent. Given our work on DU145 (this study) and the CCLE analysis of 11 tumor cell lines, it appears that RBX1 is not essential for global cell survival in a significant set of tumor cell lines. Moreover, our study has raised possibility that hemizyosity may be a significant factor that sensitizes RBX1's essentiality. Results from CRISPR experiments with the haploid KBM7 and the 17p-deficient K562 cells from the Wang paper () appear in line with our assertion. Finally, it is worth pointing out that both RBX1 and its paralog RBX2 (also named as RNF7), which shares 7 out of 8 Cys/His residues that constitute their RING finger domain, are found to be evolutionarily conserved among many species from yeast to humans (52). More importantly, either human RBX1 or RBX2 can rescue the lethal phenotype caused by deletion of the RBX ortholog Hrt1 in yeast, indicating they may functionally redundant to some extent (29)."

The above paragraph is suggested to replace authors' statements that read: "RBX1 was previously shown to be an essential gene required for mouse embryonic development (51). However, it does not appear to be essential for global cell survival. Both RBX1 and its paralog RBX2 (also named as RNF7), which shares 7 out of 8 Cys/His residues that constitute their RING finger domain, are found to be evolutionarily conserved among many species from yeast to humans (52). More importantly, either human RBX1 or RBX2 can rescue the lethal phenotype caused by deletion of the RBX ortholog Hrt1 in yeast, indicating they may functionally redundant to some extent (29)."

Response to Reviewers

Attached for consideration for publication as a Research Article in *Nature Communications* is our revised manuscript entitled, “**Heterozygous Deletion of Chromosome 17p Renders Prostate Cancer Vulnerable to Inhibition of RNA Polymerase II**”. We thank the reviewers for helpful comments and critiques. In particular, the reviewer #2 provided generous help to summarize our findings on the essentiality of RBX1 for discussion. In the revised manuscript, we have textually addressed the remaining concerns of the reviewer #2 as below.

1. Point 1: On the issue of E3 SCF (Skp2).

Reviewer’s new comments on revision 2: My first concern “the effects of Skp2 knockdown in selective cell killing are marginal (now revised Fig. S4g). They are not very different from the shNT control shown in Fig. 3a” was NOT addressed by any new data. I remain unconvinced by authors’ main hypothesis that E3 SCF (Skp2) has a significant role in the pathophysiological effects of chromosome 17p-deletion.

We completely agree with the reviewer that the SKP2 knockdown alone has relatively weaker effects on the POLR2A ubiquitination and 17p^{loss} cell proliferation when compared to those of RBX1 depletion, which we have acknowledged in the revised manuscript. In the set of experiments included in the Supplementary Fig. 4g, knockdown of SKP2 had notable better cell-killing effect on the 17p^{loss} cells in comparison with 17p^{neutral} cells. As shown from co-immunoprecipitation and mass spectrometric analyses, POLR2A/RBX1 not only interacts with CRL1 and 2 E3 ligase complexes, but also interacts with CRL3 and 7 (Supplementary Fig. 4b). Additionally, depletion of the substrate recognition subunit SKP2 could partially phenocopy the effect of RBX1 knockdown on the K63-associated POLR2A ubiquitination (Supplementary Fig. S4d). The weaker effect of SKP2, compared to that of RBX1, suggest that POLR2A may indeed interact with other CRL E3 ligases in addition to SKP2. We have revised the text to include it. We certainly do not claim that SKP2 has a **significant** role in the pathophysiological effects of chromosome 17p deletion. Instead, we think that SKP2 is one contributor among other POLR2A-interacting proteins.

2. With respect to my second concern on biochemical evidence, the response is unacceptable.

The revised Fig. S4e is a pull down experiment with crude lysates, not using ALL pure components to properly address direct protein-protein interactions. Also, the percentage of “input” was not indicated, precluding assessment of pull down efficiency. More disturbingly,

however, the authors have completely ignored my suggestion to use reconstitution experiments that would demonstrate the ability of E3 SCF (Skp2) to support the ubiquitination of POLR2A.

We apologize that we did not clearly address the reviewer's concern. We agree that reconstitution experiments would be the best way to check the RBX1-mediated ubiquitination of POLR2A. However, it is technically challenging to express and purify proteins such as POLR2A (protein size of 217KDa) using a bacterial expression system. To validate the POLR2A-SKP2 interaction, we have conducted the following additional assays:

1. Using reverse co-immunoprecipitation, we verified that POLR2A/RBX1 and SKP2 are in the same protein complex (Supplementary Fig. 4b, right panel).
2. As mentioned above, it is technically difficult to express and purify POLR2A due to its large size. We are also concerned that other components or regulatory elements in both of the RBX1-containing CRL complex and POLR2A-containing RNA Pol II complex are required for the interaction of POLR2A and SKP2. Therefore, we further confirmed the interaction of POLR2A with SKP2 using pull-down assays. We observed that purified GST-SKP2 was able to pull down a considerable amount of POLR2A when compared to the well-documented POLR2A interaction protein POLR2H, indicating the direct interaction of SKP2 with POLR2A (Supplementary Fig. 4e). The percentage of input has been indicated in the figure.
3. Using confocal single plane microscopy, we found that POLR2A co-localized significantly with RBX1 or SKP2 in the nucleus, with Pearson correlation coefficient of 0.76 and 0.44, respectively (Supplementary Fig. 4f).
4. We also found that overexpression of SKP2 significantly increased the levels of ubiquitinated POLR2A in the presence of HA-tagged ubiquitin.

Taken together, these results suggest that SKP2 recognition subunit is involved in the RBX1-mediated effect on the 17p^{loss} cancer cells through their functional interactions with POLR2A.

We agree that the reconstitution experiment will be the best way to demonstrate the E3 activity on the ubiquitination of POLR2A. To this end, we will try to isolate and purify POLR2A proteins from human cells instead of bacterial expression in the future. At this time point, we hope that the reviewer agrees that SKP2 may contribute to the RBX1-mediated ubiquitination of POLR2A.

3. I do appreciate authors' new findings in this work that may alert interested investigators to be aware of the complexity concerning RBX1 essentiality. However, the authors have not clearly

addressed this issue in the manuscript. I have prepared a paragraph that could be used by the authors in the Discussion:

“Results of CRISPR-based knockout of RBX1 (Fig. 2d) or silencing of RBX1 by shRNA (Fig. 3a, b and Supplementary Fig. 2a) in a castration-resistant prostate cancer (CRPC) cell line DU145 presented in this work have revealed that RBX1 is essential for proliferation only in 17p-deficient cells, but not in 17p-neutral parental cells. These findings suggest that RBX1 is not an essential gene for proliferation at least in DU145 cells. However, RBX1 was shown to be an essential gene required for mouse embryonic development (Tan et al, 2009, PNAS, 106: 6203-8). In addition, CRISPR-based genome-wide analysis shows that RBX1 is an essential gene in all four cell lines tested (KBM7, K562, Jiyoye, and Raji) (Wang et al., 2015, Science, 350: 1096-1101; Blomen et al., 2015, Science, 350: 1092-1095). To understand this discrepancy, we have analyzed RBX1 genetic integrity in a large set of tumor cells using Cancer Cell Line Encyclopedia (CCLE) database. The results revealed homozygous deletion of RBX1 in at least 11 cell lines, including ACHN, HT115, KALS-1, LCLC-103H, MSTO-211H, REC-1, SH-SY5Y, SK-CO-1, SK-N-SH, SNU-719, and TE-1. In addition, KBM7 (used in the Wang study) is a haploid cell line that only contains one copy of Chr17p, while K562 (used in the Wang study) is a cell line with heterozygous loss of 17p. Of note, the genomic data of Jiyoye cell line (used in the Wang study) is unavailable from CCLE. Raji cell line (used in the Wang study) seems to contain neural copy number of 17p. Collectively these data suggest that the essentiality of RBX1 for cell growth may be context-dependent. Given our work on DU145 (this study) and the CCLE analysis of 11 tumor cell lines, it appears that RBX1 is not essential for global cell survival in a significant set of tumor cell lines. Moreover, our study has raised possibility that hemizyosity may be a significant factor that sensitizes RBX1’s essentiality. Results from CRISPR experiments with the haploid KBM7 and the 17p-deficient K562 cells from the Wang paper appear in line with our assertion. Finally, it is worth pointing out that both RBX1 and its paralog RBX2 (also named as RNF7), which shares 7 out of 8 Cys/His residues that constitute their RING finger domain, are found to be evolutionarily conserved among many species from yeast to humans (52). More importantly, either human RBX1 or RBX2 can rescue the lethal phenotype caused by deletion of the RBX ortholog Hrt1 in yeast, indicating they may functionally redundant to some extent (29).”

The above paragraph is suggested to replace authors’ statements that read: “RBX1 was previously shown to be an essential gene required for mouse embryonic development (51).

However, it does not appear to be essential for global cell survival. Both RBX1 and its paralog RBX2 (also named as RNF7), which shares 7 out of 8 Cys/His residues that constitute their RING finger domain, are found to be evolutionarily conserved among many species from yeast to humans (52). More importantly, either human RBX1 or RBX2 can rescue the lethal phenotype caused by deletion of the RBX ortholog Hrt1 in yeast, indicating they may functionally redundant to some extent (29).”

We appreciate very much for the reviewer's suggestion. It is a perfect summary of our findings for discussion. We have revised our manuscript to include this paragraph.